# Risk of eczema, wheezing and respiratory tract infections in the first year of life: A systematic review of vitamin D concentrations during pregnancy and at birth

**Muzaitul Akma Mustapa Kamal Basha**[1,2☯]**, Hazreen Abdul Majid**[1,3,4☯]*****,
**Nuguelis Razali**[5‡]**, Abqariyah Yahya**[1‡]

1 Department of Social Preventive Medicine, Faculty of Medicine, University of Malaya, Kuala Lumpur, Malaysia, 2 Department of Special Care Nursing, Kulliyyah of Nursing, International Islamic University Malaysia (IIUM), Kuantan Campus, Pahang, Malaysia, 3 Department of Nutrition, Harvard Chan School of Public Health, Boston, Massachusetts, United States of America, 4 Department of Nutrition, Faculty of Public Health, University of Airlangga, Surabaya, Indonesia, 5 Department of Obstetrics & Gynaecology, Faculty of Medicine, University of Malaya, Kuala Lumpur, Malaysia

☯ These authors contributed equally to this work.
‡ These authors also contributed equally to this work.
* hazreen@ummc.edu.my

## Abstract

### Background

Allergic conditions and respiratory tract infections (RTIs) are common causes of morbidity and mortality in childhood. The relationship between vitamin D status in pregnancy (mothers), early life (infants) and health outcomes such as allergies and RTIs in infancy is unclear. To date, studies have shown conflicting results.

### Objective

This systematic review aims to gather and appraise existing evidence on the associations between serum vitamin D concentrations during pregnancy and at birth and the development of eczema, wheezing, and RTIs in infants.

### Data sources

PubMed, MEDLINE, ProQuest, Scopus, CINAHL, Cochrane Library and Academic Search Premier databases were searched systematically using specified search terms and keywords.

### Study selection

Articles on the associations between serum vitamin D concentrations during pregnancy and at birth and eczema, wheezing, and RTIs among infants (1-year-old and younger) published up to 31 March 2019 were identified, screened and retrieved.

**Data Availability Statement:** All relevant data are within the manuscript and its Supporting

Information files. The Newcastle-Ottawa Quality Assessment Form for Cohort Studies can be accessed via https://www.ncbi.nlm.nih.gov/books/NBK115843/bin/appe-fm3.pdf "

**Funding:** This review was funded by the University of Malaya Special Research Fund Assistance (BKS001-2018, under author NR) and by NAEIM (Faculty of Medicine, University of Malaya, under author HAM), primarily for article search expenses.

**Competing interests:** The authors have declared that no competing interests.

## Results

From the initial 2678 articles screened, ten met the inclusion criteria and were included in the final analysis. There were mixed and conflicting results with regards to the relationship between maternal and cord blood vitamin D concentrations and the three health outcomes— eczema, wheezing and RTIs—in infants.

## Conclusion

Current findings revealed no robust and consistent associations between vitamin D status in early life and the risk of developing eczema, wheezing and RTIs in infants. PROSPERO registration no. CRD42018093039.

## Introduction

Allergic diseases and respiratory tract infections (RTIs)–whether upper respiratory tract infections (URTIs) or lower respiratory tract infections (LRTIs)–are among the most common causes of morbidity and mortality in childhood [1, 2]. Over the years, occurrences of allergic diseases and RTIs in children have continued to rise [3–5]. The trend is apparent among the infant age group [6–8]. Worldwide, it was estimated that 30%–60% of infants suffered from allergic conditions such as eczema and wheezing [8, 9].

Previously, the prevalence of allergic diseases was most common in Europe, but in recent decades, the same phenomenon has been observed in Asia [10, 11]. A cross-sectional study in Malaysia found that the prevalence of eczema among infants aged two years and younger was 65% [12]. An earlier study reported that approximately 37% of those aged one year and younger experienced the same condition [9]. Other Asian countries such as Korea, Indonesia, Japan and Thailand reported a prevalence of eczema ranging between 5% and 23% [13–16]. On the other hand, RTIs (particularly LRTIs) are a leading cause of hospital admission among infants. A survey conducted in a Malaysian city found that 88% of hospital admissions due to LRTI involved infants aged 1 year and younger, with the median age being 8 months [17]. Likewise, 69.6% of LRTI hospital admissions in Korea involved children younger than three years, with the majority being infants [18].

The early presentation of these conditions indicates that prenatal factors such as environmental exposures during the gestational period may have a role in the development of allergic diseases in infancy [19, 20]. Among the various environmental factors that contribute to allergic diseases and RTIs, vitamin D status has gained much research interest, given its purported immuno-modulatory properties [21]. Low vitamin D levels in pregnant mothers are said to have adverse consequences on the immune system development of infants in early life. [22]. Similarly, high levels of maternal vitamin D during pregnancy were associated with lower occurrences of eczema and RTIs in infants [16, 23, 24]. In contrast, several studies did not support this association and have shown conflicting results [25, 26]. To date, the extent vitamin D levels during pregnancy and early postnatal period (e.g., cord blood) affect the development of allergic diseases and RTIs in infants remains unclear. Prior reviews were heterogeneous in their sample characteristics and methods. Some included a wide range of age groups while assessing allergic diseases and RTI outcomes (e.g. 0 to 7 years) while some employed diverse tools in the measurement of serum vitamin D and dietary intake [4, 24, 27].

The lack of conclusive empirical evidence among a more focused age group (e.g., infants aged one year and younger) has implications on health policy and clinical practice. For instance, questions of when should vitamin D be provided as a form of intervention—earlier or later—and whether it is more beneficial to prescribe vitamin D to a woman during her pregnancy or to her two-year-old child remain. In some countries, vitamin D has been recommended as supplementation for pregnant women and infants and foods were supplemented with vitamin D. This is because vitamin D deficiency is prevalent among pregnant women and infants ranging from 45% to 90% in pregnant women and 61% to 96% in infants [28]. This is likely due to inadequate sun exposure, vitamin D dietary and exclusive breast milk intake.

However, in countries that receive more sunlight (e.g., countries with equatorial climate), the necessity of vitamin D supplementation for pregnant women and infants has been questioned. Compared to previous reviews, we added to existing knowledge by summarising the data differently. We narrowed down the endpoint (age gap) to enable a clearer and more focused understanding on the impact of vitamin D levels in pregnancy and early life (cord blood) on health outcomes in the first year of life. Given the alarming trend of allergic diseases and RTIs in infancy—with vitamin D levels as a potential risk factor—we undertook a systematic review to determine the relationship between vitamin D (25(OH)D) concentrations during pregnancy and at birth (e.g., cord blood) with eczema (including atopic dermatitis), wheezing and RTIs (URTIs and LRTIs) in the first year (12 months) of life. This review focused on serum 25hydroxyvitamin D (25(OH)D) instead of vitamin D dietary intake as clinically, serum 25(OH)D represents the cumulative effect of dietary intake of vitamin D and sunlight exposure [29].

## Methods

This review was designed according to the PICOS (Participants, Intervention, Comparison, Outcomes and Study Design) criteria (Table 1) and The Preferred Reporting Items for Systematic Reviews and Meta-Analyses (PRISMA) guidelines [30] (Fig 1). The review protocol has been registered with the International Prospective Register of Systematic Reviews [PROSPERO Registration no; CRD42018093039].

### Data sources and literature search

The PubMed, MEDLINE, ProQuest, Scopus, CINAHL, the Cochrane Library and Academic Search Premier databases were used to search for studies published until March 31, 2019. The search was performed on May 1, 2019. Articles investigating associations between serum vitamin D concentrations during pregnancy and at birth (cord blood) and the development of allergic eczema, wheezing, and RTIs in infants were identified systematically. The keywords used in the search were "vitamin D" AND (wheezing OR atopic dermatitis OR eczema OR

**Table 1. PICOS criteria employed to design the systematic review.**

| Criteria | Description |
|---|---|
| Participants | Pregnant women and their infants aged ≤12months. |
| Intervention/ exposure | Serum 25(OH)D concentrations during pregnancy and early postnatal (cord blood). |
| Comparison | Analysis of serum 25(OH)D concentrations during pregnancy and early postnatal (cord blood) either by mean, median, interquartile or cut-off values such as deficiency, sufficient and insufficient. |
| Outcomes | Eczema, wheezing and RTIs of infants aged ≤12months. |
| Study design | Randomised controlled trial (RCT), non-RCT, observational cohort and case control study. |

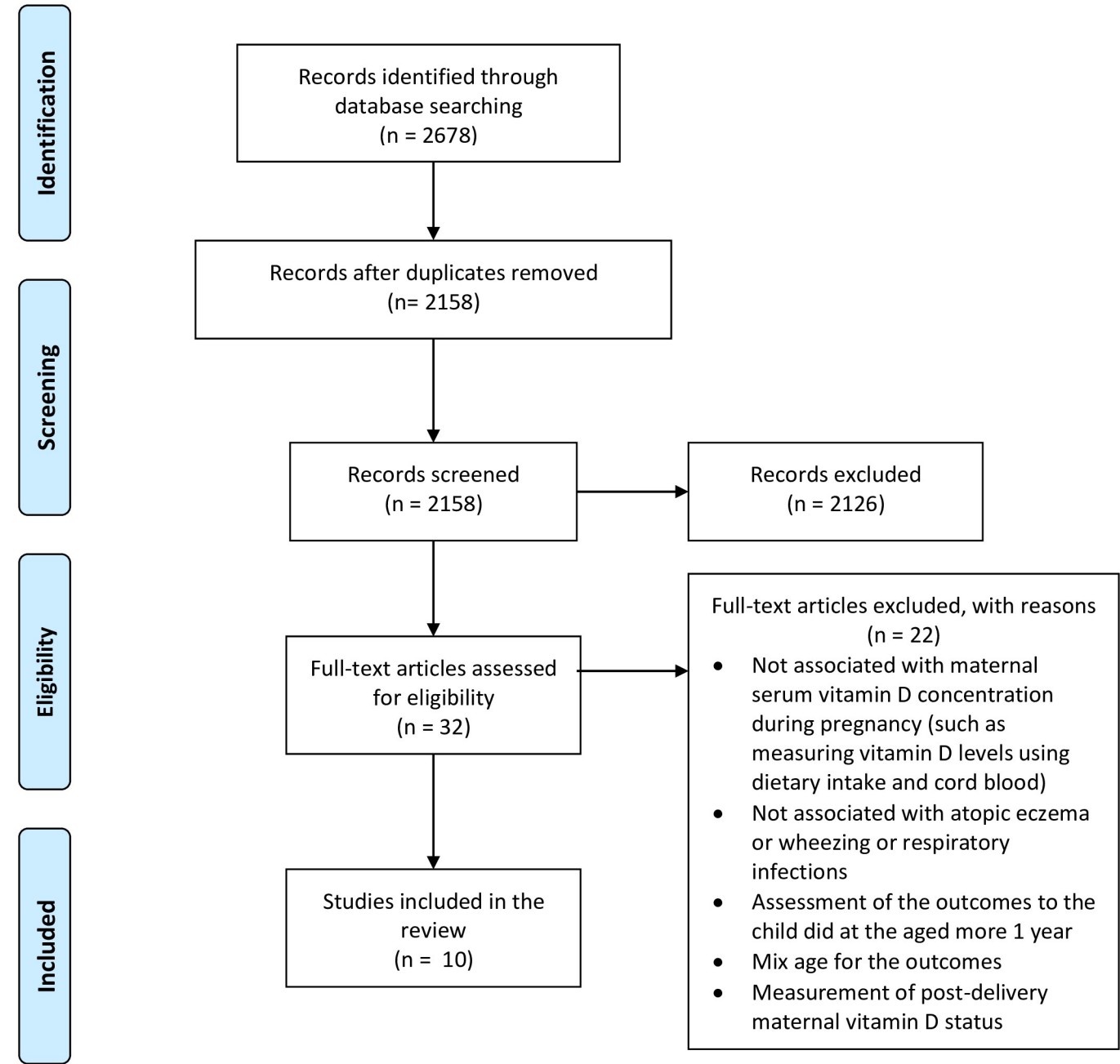

**Fig 1. PRISMA flow diagram.**

respiratory infections). In addition, all related keywords, MeSH terms, text words and search terms were used in the literature search (S1 Table).

## Study selection and data extraction

The goal of the search strategy was to identify studies that reported the associations between serum vitamin D concentrations during pregnancy and at birth (cord blood) and the development of allergic eczema (including atopic eczema/dermatitis), wheezing and RTIs (including

URTIs and LRTIs) among infants in the first year (12 months) of life. Allergic eczema, wheezing and RTIs in infants (in the 12 months of life) is defined as outcomes, which were diagnosed either by health professional officers, assessed by established questionnaires or tools or parental reports. The following inclusion criteria were applied: (1) Primary research which includes randomised controlled trial (RCT), non-RCT, observational cohort and case-control study designs, and; (2) studies that examined the associations between serum 25(OH)D concentrations during pregnancy or at birth (cord blood) and the development of allergic eczema, wheezing and RTIs among infants in the first year (12 months) of life. Meanwhile, the exclusion criteria were as follows: (1) study subjects were more than one-year old (>12 months); (2) studies that involved animals; (3) review articles, conference papers and dissertations; (4) ecological studies; (5) non-English papers, and; (6) studies published as only abstracts without full-texts.

The database search identified titles and abstracts that were independently screened and reviewed by two authors. Results were then compared, and disagreements were resolved by discussion and by consulting a third author when necessary. The following information was extracted from the selected papers: Author (s); year of publication; study population; sample size; study location; study design; parental allergy history; exposure; week of pregnancy during measurement of vitamin D, how the outcomes were assessed, which either (1) were diagnosed by health professional officers (including parental reports); (2) assessed by established questionnaires or tools such as 'SCORing Atopic Dermatitis' (SCORAD) index or (3) parental reports (based on parent observation or perception); and summary of the association for exposure and results of the outcomes (eczema, wheezing and respiratory infections). The original authors of the papers were contacted whenever further clarification of the study results or other details were required.

## Assessment of risk of bias

The Cochrane Collaboration's tool was used to assess the risk of bias in randomised control trials [31]. Meanwhile, the Eight-Star Newcastle-Ottawa Scale (NOS) for Cohort studies was used to assess the risk of bias in observational studies [32] (S2 Table).

## Data synthesis and analysis

Mean, median, odd ratios (OR) and relative risks (RR) of the outcomes were regarded as primary measures of exposure and outcomes effect. However, other calculations/statistics, such as quartiles were presented in this review. Final results and scoring of selected papers were tabulated, summarised and elaborated in a narrative manner.

## Results

A total of 2678 articles were initially identified from the databases. Following title and abstract screening, 32 full-texts were analysed. Out of 32 studies, ten met the inclusion criteria. Twenty-two were then removed because the outcome measurement was not specific to those one-year-old and younger, or it was performed on subjects older than one year. Fig 1 illustrates the flow diagram of literature search (PRISMA).

## Study characteristics

All the ten papers selected were prospective cohort studies conducted in western, high-income nations [7, 33–41]. Some studies assessed multiple outcomes. Six studies measured the associations between vitamin D concentrations in pregnancy and allergic eczema, five measured

wheezing as the outcome and six measured RTIs. Methods of outcome assessment were diverse. Most studies relied on parental reports (n = 7) while some used standardised tools (n = 3) to determine eczema: 'SCORing Atopic Dermatitis' (SCORAD) index [7], 'UK Working Party's diagnostic criteria' [33] and 'Hanifin and Rajka criteria' [41]. Six studies measured maternal serum 25(OH)D concentrations, while four studies measured cord serum 25(OH)D as their exposure component. In terms of measurement timing, one study conducted its assessment in the stage of early pregnancy (first trimester) [34], another in mid-pregnancy (second trimester) [38] and four studies in late pregnancy (third trimester) [33, 35–37].

A further five studies reported the ethnic backgrounds of their study subjects [7, 33, 34, 40, 41]. For instance, in Gale et al. (2008), all participants were Caucasians, and in Morales et al. (2011), 97% were Caucasians. Jone et al. (2012), Camargo et al. (2010) and Palmer et al. (2015) included various ethnic groups but Caucasians still comprised the majority of the study subjects. Meanwhile, age of subjects ranged between 0 to 12 months when outcomes (wheezing, allergic eczema or RTIs) were measured. Most studies reported either odd ratios (OR) or relative risks (RR) with the 95% confidence interval except for one study that reported only p-values [36]. Detailed descriptions of the ten studies are presented in Table 2.

## Quality assessment

Of the ten cohort studies, nine could be regarded as good (high) quality [7, 33–35, 37–41], and one was scored to be of moderate or fair quality [36]. Details summary of quality appraisal is illustrated in Table 3.

## Vitamin D status

The average concentration of serum 25(OH)D during pregnancy ranged between 50.0 to 73.6 nmol/L, while in the early postnatal period (cord blood) it ranged between 44.0 to 58.4 nmol/L. This might be due to the differences in the timing of exposure assessment and methods of reporting. Six studies reported median with interquartile range, with the quartiles and quintiles being either natural or pre-set [33–35, 37, 39, 40]. Three studies reported mean and standard deviation for the serum 25(OH)D levels [7, 38, 41]. One study categorised serum 25(OH)D levels into three groups: deficient ($<$ 50 nmol/L), insufficient (50.0–75.0 nmol/L) and sufficient ($>$ 75.0 nmol/L). To determine the associations between exposure (vitamin D concentrations during pregnancy or at birth) and outcomes (eczema, wheezing or RTIs), some studies used quartiles and quintile while some employed cut-off values while categorising vitamin D levels, such as deficient (25.0–40.0, $<$ 50.0 nmol/L), sufficient (50.0–74.9, $>$75.0 nmol/L), severe deficiency ($\leq$25.0 nmol/L), insufficient (50.0–75.0 nmol/L) and optimal ($\geq$75.0 nmol/L). These variations rendered comparison across studies difficult. In addition, there has been no consensus on the threshold value for an optimal level of vitamin D.

## Vitamin D and eczema in the first year (12 months) of life

Six studies examined maternal or cord serum 25(OH)D levels in relation to eczema in infants aged one and younger. Three studies reported on the association during pregnancy, while the other three studies reported in cord blood [7, 33, 35, 38, 39, 41]. All were ranked as high quality based on the NOS (refer Table 4). Out of the three studies that assessed vitamin D levels in pregnancy, two had their measurement conducted in late pregnancy (28–42 weeks of gestation) and gave conflicting results [33, 35]. Gale et al. (2008) reported that higher maternal serum 25(OH)D concentrations in late of pregnancy led to higher risks of eczema in 9-month-old infants (Adjusted OR = 3.26, 95% CI = 1.15–9.29). Whereas, Weisse et al. (2013) found no significant association between maternal serum 25(OH)D concentrations in late pregnancy

**Table 2. Characteristics of studies included in the final analysis (n = 10).**

| Author | Study setting | Study design, participant | Exposure | Parental allergy/ atopy history | Period of exposure assessment | Outcomes | How the outcome was assessed | Age during outcome assessment |
|---|---|---|---|---|---|---|---|---|
| Gazibara[a], 2016 [38] | Rotterdam, Netherlands (51°55'N) | Cohort study 3019 mother–child pairs | Maternal 25(OH)D serum | Reported | 2nd trimester Median of gestation = 20.5 (IQR: 18.1–24.9) weeks | • Eczema | • Parent reported | • At 6 months (mo) <br> • At 12 mo |
| Palmer, 2015 [41] | Adelaide, Australia (34° 51' S) | Cohort study 270 mother–child pairs | 25(OH)D cord blood | Reported | Delivery | • RTIs <br> • Atopic eczema | • Parent reported <br> • Hanifin & Rajka criteria | • At 12 mo |
| Baiz, 2014 [39] | Pontiers (46° 35'N) Nancy (48° 41'N), France | Cohort study 2002 mother–child pairs | 25(OH)D cord blood | Reported | Delivery | • Wheezing <br> • Atopic eczema | • Parent reported <br> • Parent reported | • At 12 mo |
| De Jongh, 2014 [37] | Southampton, UK, (50° 54'N) | Cohort study 2025 mother–child pairs | Maternal 25(OH)D serum | Not reported | At week 34 of gestation | • Wheezing <br> • URTIs <br> • LRTIs | • Parent reported <br> • Parent reported <br> • Parent reported | • At 6 mo <br> • At 12 mo |
| Skowrońska-Jóźwiak, 2014 [36] | Poland | Cohort study 102 mother–child pairs | Maternal 25(OH)D serum | Not reported | 3rd trimester | • RTIs | • Parent reported | • At 12 mo |
| Weisse, 2013 [35] | Leipzig, Germany (51° 40'N) | Cohort study 378 mother–child pairs Ethnicity: not reported | Maternal 25(OH)D serum | Reported | 3rd trimester At week 34 of gestation | • Atopic eczema | • Parent reported | • At 12 mo |
| Jones, 2012 [7] | Perth, Australia (31°57'S) | Cohort study 231 mother–child pairs | 25(OH)D cord blood | Reported | Delivery | • Wheezing <br> • Atopic eczema | • Parent reported <br> • SCORAD severity index | • At 12 mo |
| Morales, 2012 [34] | 4 study areas: Menorca Island, Valencia, Sabadell and Gipuz-koa in Spain (39°N, 39°N, 41°N, 44°N) | Cohort study 1724 mother–child pairs Ethnicity; 97% white, 3% others | Maternal 25(OH)D serum | Reported | Early pregnancy Mean of gestation = 12.6 (SD ±2.5) weeks | • LRTIs <br> • Wheezing | • Parent reported <br> • Parent reported | • At 12 mo |
| Camargo, 2010 [40] | Wellington (41°S), Christchurch (43°S), New Zealand | Cohort study 922 newborns | 25(OH)D cord blood | Not reported | Delivery | • RTIs | • Parent reported | • At 3 mo |
| Gale, 2008 [33] | Southampton, UK (50° 54'N) | Cohort study 596 mother–child pairs Ethnicity: 100% white | Maternal 25(OH)D serum | Not reported | 3rd trimester Median of gestation = 32.6 (IQR: 32.0–33.4) weeks | • Atopic eczema <br> • RTIs | • UK Working Party's diagnostic criteria <br> • Parent reported | • At 9 mo |

and eczema in infants aged 0–1 year (Adjusted OR = 0.89, 95% CI = 0.63–1.32, P-value = 0.614). Nevertheless, Weisse et al. (2013) adjusted for parental allergy history in the analyses but the assessment of the eczema relied on parental report. Whereas Gale et al. (2008) did not report they had adjusted for parental allergy history, but they used the established tool to assess the eczema outcomes. On the other hand, a study conducted by Gazibara et al. (2016) in mid-gestation (median 20.5 (IQR, 18.1–24.9) weeks of gestation) reported that the 25(OH)D levels were not associated with eczema in infants (parental allergy history was accounted in the analyses, and the eczema outcome was based on parental reports).

**Table 3. Summary of quality assessment of the included studies using the Newcastle-Ottawa scale.**

| | Selection | | | | Comparability | Outcome | | |
|---|---|---|---|---|---|---|---|---|
| | Representativeness of the exposed cohort | Selection of the non-exposed | Ascertainment of exposure | Demonstration that outcome of interest was not present at start of study | Comparability of cohorts on the basis of the design or analysis controlled for confounders | Assessment outcome | Follow up long enough for outcomes to occur | Adequacy of follow up of cohorts |
| Palmer et al. (2015) | * | * | * | * | * | * | * | * |
| Jones et al. (2012) | * | * | * | * | * | * | * | * |
| Baiz et al. (2014) | * | * | * | * | * | | * | * |
| Camargo et al. (2010) | * | * | * | * | * | | * | * |
| Weisse et al. (2013) | * | * | * | * | * | | * | |
| Gale et al. (2008) | * | * | * | * | | * | * | * |
| Gazibara et al. (2016) | * | | * | * | * | | * | |
| De Jongh et al. (2014) | * | | * | * | * | | * | |
| Morales et al. (2012) | * | | * | * | * | | * | |
| Skowrońska-Jóźwiak et al. (2014) | * | | * | * | | | * | |

**Good quality**: 3 or 4 stars in selection domain AND 1 or 2 stars in comparability domain AND 2 or 3 stars in outcome/exposure domain.

**Fair quality**: 2 stars in selection domain AND 1 or 2 stars in comparability domain AND 2 or 3 stars in outcome/exposure domain.

**Poor quality**: 0 or 1 star in selection domain OR 0 stars in comparability domain OR 0 or 1 stars in outcome/exposure domain.

Further, three studies addressed the association between cord blood 25(OH)D concentrations and the eczema outcome in the 12 months of life in infants [7, 39, 41]. Two studies reported that cord blood 25(OH)D concentrations were significantly contributed to the risk of eczema in the first 12 months of life in infants [7,41]. Both studies, Jones et al. (2012) and Palmer et al. (2015), used the established tool to assess the eczema outcomes. In contrast, Baiz et al. (2014) defined the outcome of eczema based on parental report did not find a significant association between cord blood 25(OH)D concentrations and the eczema outcome in the 12 months of life in infants. However, these three studies adjusted for family allergy history in their analyses [39, 41]. Detailed descriptions are available in Tables 2 and 4.

## Vitamin D and wheezing in the first year (12 months) of life

Four studies reported the association between serum 25(OH)D concentrations and wheezing in the first year of life. They found that the wheezing outcome for all studies were based on parental reports. Two assessed 25(OH)D levels during pregnancy and two assessed 25(OH)D levels in cord blood [7, 34, 37, 39]. All were ranked as good (high) quality based on the NOS (Table 3). Morales et al. (2011) studied 1724 mother–child pairs in Spain and found no association between maternal serum 25(OH)D in early pregnancy and wheezing at the age of 1 year (P trend = 0.441). This was in line with Jone et al.'s (2012) findings using cord blood. On the contrary, Baiz et al. (2014) demonstrated that higher vitamin D concentrations in cord blood

**Table 4. Studies included in the analysis of 25 (OH)D during pregnancy or cord blood with the development of eczema in the first year (12 months) of life.**

| Author | serum 25(OH) D levels | Result findings | Conclusion |
|---|---|---|---|
| Gazibara[a], 2016 [38] | Mean (SD) maternal 25(OH) D level = 65.5 (43.7) nmol/L | Eczema (0–1 y); N = 614 | No association between maternal serum 25(OH)D levels in 2nd trimester and the risk of eczema in the first year of life. |
| | | OR (95% CI) | |
| | Severely deficient = <25.0 nmol/L | <25.0 nmol/L: 0.91 (0.55–1.49) | |
| | Deficient = 25.0–49.9 nmol/L | 25.0–49.9 nmol/L: 0.98 (0.74–1.31) | |
| | Sufficient = 50.0–74.9 nmol/L | 50.0–74.9 nmol/L: 0.92 (0.72–1.17) | |
| | Optimal ≥75.0 nmol/L | ≥75.0 nmol/L: reference | |
| | | P for trend: 0.940 | |
| Palmer[b], 2015 [41] | Mean (SD) 25(OH)D cord blood level = 57.0 (24.1) nmol/L | Eczema at 1 year: N = 265 | Risk of eczema at 1 year of age decreased (12% reduction in risk) as cord blood 25(OH)D concentrations increased; a 10 nmol/L rise. |
| | | Adjusted RR (95% CI) in relation to 10 nmol/L rise 25(OH)D cord blood: **0.88 (0.81–0.96)** | |
| | | *P*-value = **0.002** | |
| Baiz[c], 2014 [39] | Median (IQR) 25(OH)D cord blood level = 17.8 (15.1) ng/ml[+] | Atopic eczema at 1 year: N = 239 | No association between cord blood 25(OH)D levels and the risk of eczema in the first year of life. |
| | | Adjusted OR (95% CI) in relation to 5 ng/ml[+] rise 25(OH)D cord blood = 0.84 (0.71–1.00) | |
| | | *P*-value = 0.050 | |
| Weisse[d], 2013 [35] | Median (IQR) maternal 25(OH)D level = 22.19 (14.40–31.19) ng/ml[+] | Atopic eczema (0–1 y) | No association between maternal serum 25(OH)D levels in 3rd trimester and atopic eczema in the first year of life. |
| | | Symptoms (N = 272) | |
| | Q1: 6.13–14.39 ng/ml[+] | Maternal 25(OH) D, quartiles: n (%) | |
| | Q2: 14.40–22.19 ng/ml[+] | Q1: 10 (15.2) | |
| | Q3: 22.20–32.19 ng/ml[+] | Q2: 8 (11.0) | |
| | Q4: 32.20–60.80 ng/ml[+] | Q3: 6 (9.5) | |
| | | Q4: 7 (10.0) | |
| | | Adjusted OR (95% CI) 0.89 (0.63–1.32); P = 0.614 | |
| | | Diagnosed (N = 272) | |
| | | Maternal 25(OH) D, quartiles: n (%) | |
| | | Q1: 4 (6.1) | |
| | | Q2: 9 (12.3) | |
| | | Q3: 7 (11.1) | |
| | | Q4: 7 (10.0) | |
| | | Adjusted OR (95% CI): 1.16 (0.79–1.71); P = 0.451 | |
| Jones[e], 2012 [7] | Mean (SD) 25(OH)D cord blood level = 58.4(SD, 24.1) nmol/L | Eczema at 1 year; N = 193 | Risk of eczema at 1 year of age decreased (14% reduction in risk) as cord blood 5(OH)D concentrations increased; a 10 nmol/L rise. |
| | | Adjusted OR (95% CI) in relation to 10 nmol/L rise 25(OH)D cord blood: **0.86 (0.74–0.99)** | |
| | | *P*-value = **0.042** | |
| Gale[NA], 2008 [33] | Median (IQR) maternal 25(OH)D level = 50.0 (IQR: 30.0–75.3) nmol/L | Eczema (0–9 mo); OR (95%CI) N = 440 | High maternal serum 25(OH)D level (>75 nmol/l) in 3rd trimester have a higher risk of atopic eczema in the first year of life. |
| | Q1: < 30 nmol/L (reference) | Q1: 1.0 (reference) | |
| | | Q2: 0.59 (0.14–2.50) | |
| | Q2: 30–50 nmol/L | Q3: 0.79 (0.21–3.00) | |
| | Q3: 51–75 nmol/L | **Q4: 3.26 (1.15–9.29)** | |
| | Q4: > 75nmol/L | | |

[a] Adjusted for maternal age, maternal pre-pregnancy body mass index, maternal education level, maternal history of eczema, allergy and asthma, parity, smoking, pet keeping, psychological distress, folate level, seasonal, gestational age at birth, birth weight, breastfeeding, vitamin D supplementation at the age 2 months.

[b] Adjusted for parity, gender, maternal smoking, and vitamin D supplementation.

[c] Adjusted for city, mother's age, maternal history of allergy, pre-pregnancy body mass index, any smoking during pregnancy, any passive smoke exposure during the first 3 years of life, number of siblings, household income, newborn's sex and weight, season of birth, and exclusive breast-feeding for 4 months or longer.

[d] Adjusted for child sex, number of siblings, cord blood total IgE, family history of atopy, breastfeeding, UV intensity of month of birth and vitamin D supplementation during the 1st year of life.

[e] Adjusted for season of birth, infant gender, pet ownership, maternal age and maternal ethnicity.

[NA] Not reported for adjusted cofounder.

[+] 1ng/ml is equivalent 2.496 nmol/L.

N = study size n = number of cases.

**Table 5. Studies included in the analysis of 25 (OH)D during pregnancy or cord blood with the development of wheezing in the first year (12 months) of life.**

| Author | Finding of serum 25(OH) D levels | Finding of the association | Conclusion |
|---|---|---|---|
| Baiz[c], 2014 [39] | Median (IQR) 25(OH)D cord blood level = 17.8 (IQR 15.1) ng/ml[+] | Wheezing at 1 year: N = 239 | High levels cord blood 25(OH)D levels decreased the risk of wheezing at the age of 1 year. |
| | | Adjusted OR (95% CI) in relation to 5 ng/ml[+] rise 25(OH)D cord blood: = **0.67 (0.54–0.81)** | |
| | | *P*-value < **0.001** | |
| De Jongh[f], 2014 [37] | Median (IQR) maternal 25(OH)D level = 59.0 (40.6–84.3) nmol/L | RR (95% CI) | Low maternal serum 25(OH)D levels decreased the risk of wheezing at 0–6 months but not at 6–12 months. |
| | Q1: < 25 nmol/L | Wheezing (0–6 mo) n/N = 525/2021 | |
| | Q2: 25–49 nmol/L | **Q1: 0.64 (0.44–0.95)** | |
| | Q3: 50–74 nmol/L | **Q2: 0.72 (0.61–0.87)** | |
| | Q4: ≥75nmol/L (reference) | Q3: 0.96 (0.81–1.15) | |
| | | Q4: Reference | |
| | | P = **0.000** | |
| | | Wheezing (6–12 mo) n/N = 601/1946 | |
| | | Q1: 1.10 (0.80–1.52) | |
| | | Q2: 1.21 (1.03–1.43) | |
| | | Q3: 1.17 (0.98–1.39) | |
| | | Q4: Reference | |
| | | P = 0.163 | |
| Jones[e], 2012 [7] | Mean (SD) 25(OH)D cord blood level = 58.4 (SD, 24.1) nmol/L | Wheezing at 1 year: N = 231 | No risk of wheezing related to cord blood 25(OH)D concentrations at the age of 1 year. |
| | | Adjusted OR (95% CI) in relation to 10 nmol/L rise 25(OH)D cord blood: | |
| | | 1.00 (0.98–1.01) | |
| | | *P*-value = 0.731 | |
| Morales[g], 2012 [34] | Median (IQR) maternal 25(OH)D level = 29.5 (22.5–37.1) ng/ml[+] | Wheezing (0–1 y); (N = 1724) | No association between maternal serum 25(OH)D levels in early of pregnancy and the risk of wheezing in the first year of life. |
| | Q1: < 21.9 ng/ml[+] (reference) | Adjusted OR (95% CI) | |
| | | Q1: 1.0 (reference) | |
| | Q2: 21.9–29.1 ng/ml[+] | Q2: 1.04 (0.78–1.40) | |
| | Q3: 29.2–37.0 ng/ml[+] | Q3: 0.96 (0.71–1.29) | |
| | Q4: >37.0 ng/ml[+] | Q4: 0.91 (0.67–1.23) | |
| | | P trend = 0.441 | |

[c] Adjusted for city, mother's age, maternal history of allergy, pre-pregnancy body mass index, any smoking during pregnancy, any passive smoke exposure during the first 3 years of life, number of siblings, household income, newborn's sex and weight, season of birth, and exclusive breast-feeding for 4 months or longer.

[f] Adjusted for child sex, birth weight, and gestational age, maternal age, maternal education level, parity, ethnicity, breastfeeding duration, maternal smoking in pregnancy and maternal pre-pregnancy body mass index.

[e] Adjusted for season of birth, infant gender, pet ownership, maternal age and maternal ethnicity.

[g] Adjusted for child sex, maternal education level, siblings at birth, breastfeeding duration, maternal smoking in pregnancy, day-care attendance, maternal asthma and maternal pre-pregnancy body mass index.

[+] 1ng/ml is equivalent 2.496 nmol/L.

N = study size; n = number of cases.

reduced the risk of wheezing in infants. De Jongh et al. (2014) assessed wheezing at two stages (0–6 months and 6–12 months). The risk of wheezing was reduced by 36% when mothers had less than 25.0 nmol/L of vitamin D concentrations (RR = 0.64 95% CI = 0.44–0.95; P = <0.0001). This was applied to 0-6-month-old infants, while no association was found among those aged 6–12 months. Detailed descriptions are available in Table 5.

## Vitamin D and respiratory tract infections in the first year (12 months) of life

Six articles addressed the relationships between serum 25(OH)D concentrations and respiratory infections in the first year (12 months) of life. Four studies assessed 25(OH)D during pregnancy, and two assessed 25(OH)D in cord blood [33, 34, 36, 37, 40, 41]. Among the four studies that measured 25(OH)D during pregnancy, one assessed early pregnancy [34], and the rest in late pregnancy (28–42 weeks of gestation) [33, 36, 37]. Overall, the findings showed inconsistent results. Three studies did not show any association between vitamin D levels and respiratory infections [33, 37, 41]. However, Gale et al. (2008) found that children whose mothers were in the top quarter (of vitamin D level) were more likely to have been diagnosed with pneumonia or bronchiolitis than those whose mothers were in the bottom quarter (OR = 4.80, 95% CI = 1.01–22.73). In contrast, De Jongh et al. (2014) reported that mothers with serum 25(OH)D less than 50 nmol/L reported fewer lower respiratory infections in their children aged 0–6 months compared to mothers with serum 25(OH)D level more than 75nmol/L. However, the result may be attributable to residual confounding [37]. Nevertheless, both studies were ranked as 'high quality' using the NOS [33, 37].

The remaining three studies demonstrated associations between serum 25(OH)D concentrations and risks of respiratory infections in the first year of life [34, 37, 40]. Morales et al. (2011) reported that higher circulating maternal 25(OH)D concentrations in early pregnancy led to lower risks of LRTI. This corroborated the findings of Camargo et al. (2011) which measured cord blood 25(OH)D levels. Similarly, Skowronska-Jozwiak et al. (2014) found that vitamin D deficiency in late pregnancy increased the risks of respiratory infections (P = 0.003). Detailed descriptions are available in Table 6.

## Discussion

This systematic review identified ten studies examining the associations between vitamin D concentrations (during pregnancy and at birth), allergic conditions (i.e., eczema and wheezing) and RTIs in the first year of life. Overall findings demonstrated no consistent associations, as results were mixed. The amount of existing research and evidence is rather limited, and comparisons across studies remains a challenge given the different timing and tools of assessment.

### Vitamin D status

All the studies included in this review were conducted in high-income Western countries. None were from Asian countries. Three studies reported that the average vitamin D concentrations in pregnancy or at birth were deficient, whereas the remaining studies (seven studies) reported insufficient and sufficient (the majority were in the insufficient group). Serum vitamin D is preferable to determine individuals' vitamin D levels instead of dietary vitamin intake alone as it represents the cumulative effect of vitamin D dietary intake and sun exposure [29, 42].

Further, many studies reported that populations living with insufficient sunshine either most of the year or at least one month (higher latitudes; western countries) have a high risk of vitamin D deficiency [43–45]. This deficiency is probably due to inadequate sun exposure; a major source of vitamin D. Nevertheless, studies in Asian countries (have sun exposure most of the time) reported a high prevalence of vitamin D deficiency in all age groups including children and pregnant women [46–50]. The high prevalence of vitamin D deficiency in Asian

**Table 6. Studies included in the analysis of 25 (OH)D during pregnancy or cord blood with the development of RTIs in the first year (12 months) of life.**

| Author | Finding of serum 25(OH) D levels | Finding of the association | Conclusion |
|---|---|---|---|
| Palmer[b], 2015[41] | Mean (SD) 25(OH)D cord blood level = 57.0 (24.1) nmol/L | RTIs at 1 year (n/N = 45/267) | No association between 25(OH)D cord blood levels and RTIs at the age of 1 year. |
| | | Adjusted RR (95% CI) in relation to 10nmol/L rise 25(OH)D cord blood = 1.07 (0.97–1.18) | |
| | | *P*-value = 0.18 | |
| Skowrońska-Jóźwiak[NA], 2014 [36] | Mean (SD) maternal 25(OH)D level = Not reported | RTIs (0–1 y) (N = 102) | Maternal 25(OH)D deficiency in 3rd trimester may increase the risk of respiratory infections in the first year of life. |
| | Deficient = <20.0 ng/ml[+] | >30.0 ng/ml[+] vs <20.0 ng/ml[+] | |
| | Insufficient = 20.0–30.0 ng/ml[+] | P = **0.003** | |
| | Sufficient > 30.0 ng/ml[+] | >30.0 ng/ml[+] vs 20.0–30.0 ng/ml[+] | |
| | | P = **0.004** | |
| | | 20.0–30.0 ng/ml[+] vs < 20.0 ng/ml[+] | |
| | | P = 0.200 | |
| De Jongh[f], 2014 [37] | Median (IQR) maternal 25(OH)D level = 59.0 (40.6–84.3) nmol/L | LRTI (0–6 mo) | Low maternal serum 25(OH)D levels decreased the risk of LRTI at 0–6 months but not at 6-12months. |
| | | n/N = 288/2021 | |
| | Q1: < 25 nmol/L | **Q1: 0.45 (0.24–0.84)** | |
| | Q2: 25–49 nmol/L | **Q2: 0.63 (0.49–0.81)** | |
| | Q3: 50–74 nmol/L | **Q3: 0.76 (0.58–0.99)** | |
| | Q4: ≥75nmol/L (reference) | Q4: Reference | |
| | | P = **0.000** | |
| | | LRTI (6–12 mo) n/N = 368/1946 | |
| | | Q1: 1.11 (0.72–1.71) | |
| | | Q2: 1.22 (0.97–1.54) | |
| | | Q3: 1.12 (0.87–1.42) | |
| | | Q4: Reference | |
| | | P = 0.155 | |
| Morales[g], 2012 [34] | Median (IQR) maternal 25(OH)D level = 29.5 (22.5–37.1) ng/ml[+] | LRTI (0–1 y) (N = 1693) | Higher maternal 25(OH)D concentrations in early of pregnancy decreased the risk of LRTI in the first year of life. |
| | Q1: < 21.9 ng/ml[+] (reference) | Adjusted OR (95% CI) | |
| | | Q1: 1.0 (reference) | |
| | Q2: 21.9–29.1 ng/ml[+] | Q2: 0.89 (0.67–1.19) | |
| | Q3: 29.2–37.0 ng/ml[+] | Q3: 0.92 (0.70–1.23) | |
| | Q4: >37.0 ng/ml[+] | **Q4: 0.67 (0.50–0.90)** | |
| | | P trend: **0.016** | |
| Camargo[h], 2010 [40] | Median 25(OH)D cord blood (IQR) nmol/L | RTIs (0–1 y); n = 553 | High levels 25(OH)D cord blood decreased the risk of RTIs among infants in the first year of life. |
| | N = 922: 44 (29–78) nmol/L | OR (95% CI) | |
| | n = 180; <25.0 nmol/L: 19 (14–22) nmol/L | <25.0 nmol/L: **2.04 (1.13–3.67)** | |
| | n = 491; 25.0–74.0 nmol/L: 41 (34–53) nmol/L | 25.0–74.0 nmol/L: **2.16 (1.35–3.46)** | |
| | | ≥75.0 nmol/L: reference | |
| | n = 251≥75.0 nmol/L: 100 (87–124) nmol/L | P for trend: **0.030** | |

*(Continued)*

**Table 6.** (Continued)

| Author | Finding of serum 25(OH) D levels | Finding of the association | Conclusion |
|---|---|---|---|
| Gale[NA], 2008 [33] | Median (IQR) maternal 25(OH)D level = 50.0 (IQR: 30.0–75.3) nmol/L | LRTIs (pneumonia /Bronchiolitis); | Higher maternal serum 25(OH)D levels increased the risk of LRTIs BUT no association was seen in RTIs in the first year of life. |
| | | (0–9 mo) N = 440 | |
| | Q1: < 30 nmol/L (reference) | OR (95%CI) | |
| | Q2: 30–50 nmol/L | **Q4: 4.80 (1.01–22.73)** | |
| | Q3: 51–75 nmol/L | Other quarters not reported. | |
| | Q4: > 75nmol/L | Chest infections, Bronchitis/respiratory infections—reported no significant association—result not shown. | |

[b] Adjusted for parity, gender, maternal smoking, and vitamin D supplementation.

[NA] Not reported for adjusted cofounder.

[f] Adjusted for child sex, birth weight, and gestational age, maternal age, maternal education level, parity, ethnicity, breastfeeding duration, maternal smoking in pregnancy and maternal pre-pregnancy body mass index.

[g] Adjusted for child sex, maternal education level, siblings at birth, breastfeeding duration, maternal smoking in pregnancy, day-care attendance, maternal asthma and maternal pre-pregnancy body mass index.

[+] 1ng/ml is equivalent 2.496 nmol/L.

[h] Adjusted for season of birth plus 14 potential confounders (study site, maternal age at birth, New Zealand Deprivation Index, maternal history of asthma, paternal history of asthma, gestational age, gender, child's ethnicity, any smoking during pregnancy, any passive smoke exposure at 3 months of age, number of children younger 16 years in household at the time the child was 3 months old, endotoxin on bedroom floor at the time the child was 3 months old [in quartiles], damp musty smell in any room of home at the time the child was 3 months old, and duration of exclusive breastfeeding). [NA] Not reported for adjusted cofounder. N = study size; n = number of cases.

countries was probably due to clothing styles (wearing a full dress) and ethnicity [46]. These portrayed that the vitamin D concentrations are a biomarker of sun exposure rather than being the active molecule that is being studied. Although other molecules activated by the sun may confer benefits, the literature on this is scarce probably due to difficulties in analyses as it depends on the scale of the study, budget and ethical considerations. Also, there is a lack of studies in differentiating active and non-active molecules of vitamin D.

## Association between serum vitamin D concentrations in pregnancy and eczema in the first year (12 months) of life

As mentioned earlier, the comparison of results across studies in this review was difficult. Among the reasons were differences in the timing of vitamin D measurement and the lack of adjustment for similar confounders, such as parental allergy history. Throughout pregnancy, vitamin D levels could fluctuate as a result of various factors (e.g., morning sickness, limited outdoor activity and supplement intake) [43]. Therefore, vitamin D levels measured at a single period point (e.g., first trimester) may not represent the actual serum pregnancy level. It is thus difficult to ascertain whether optimal vitamin D level in pregnancy for the investigated outcomes is a risk or vice versa. On the other hand, parental allergy history should be taken into account during analysis because the tendency to develop allergies is often genetic, with heritability estimates varying between 71% and 84% [51–53]. Although genome-wide association studies (GWAS) and meta-analyses of GWAS have begun to shed light on both common and distinct pathways that contribute to allergic diseases, genetic components were acknowledged to play a role in the pathophysiology of allergic eczema [54, 55].

## Association between serum vitamin D concentrations in pregnancy and wheezing in the first year (12 months) of life

Our findings showed a lack of consistent data addressing the association between serum vitamin D concentrations in pregnancy and wheezing in the first year (12 months) of life. This contradicted a recent meta-analysis that reported no association between early life vitamin D levels and the risk of wheezing in later life [4]. Nevertheless, a number of earlier studies suggested that higher vitamin D levels in pregnancy lower the risk of wheezing during childhood [24, 27, 56]. In comparison, our review showed inconsistent findings. The potential explanation for this discrepancy, first, could be due to the differences in sample age; we focused on infants aged one year and younger while prior reviews included children between 0–7 years in which the huge age scale may dilute some information subsequently affected the outcome findings among infants population. Second, the serum 25(OH)D levels in studies were measured at different times points and measured only once. The inconsistency in the measurement time point might introduce errors in exposure assessment, but out of 4, 3 studies in this review measured at late pregnancy (at 34 weeks of gestation and birth (cord blood), thus the measurement timing should not substantially bias our findings. Nevertheless, measured 25(OH)D levels only once might not adequately reflect the long-term exposure and might attenuate any possible association of not only wheezing but also other interest in this study (eczema and RTIs) in the first year (12 months) of an infant's life.

## Association between serum vitamin D concentrations and respiratory tract infections in the first year (12 months) of life

Earlier systematic reviews indicated a protective role of vitamin D in early life against RTIs, where inverse associations were found between vitamin D levels and RTIs among infants [24, 27]. However, current findings did not support this due to several possible explanations. Firstly, the current review focused on infants aged 0–1 year, while prior reviews included a wider age-range (e.g. 0 to 7 years). In order to know whether it is more beneficial to prescribe vitamin D earlier (e.g. during pregnancy) or later (e.g. postnatal period or infants) as a form of intervention, a more focused age group (e.g. infants one year and younger) is needed. This evidence seems to be important as it has implications on health policy and clinical practice. Secondly, the outcome of RTIs in all studies was reported by parents in which the likelihood of misclassification could occur for example parents might report wheezing as respiratory infections while wheeze can also be associated with allergic conditions [57].

## Strengths and limitations of this review

This review has several strengths. First, it was conducted according to the Preferred Reporting Items for Systematic Reviews and Meta-Analyses (PRISMA) guidelines. Second, most studies were rated as good (high) quality—except for one—based on the NOS assessment for cohort studies. Third, our review focused on children aged 12 months and younger. By narrowing down the age range, we reduced the discrepancies of magnitude and latitude of other variables which may affect the outcomes. Accordingly, it could also more closely reflect the importance of vitamin D in utero exposure on the extra-skeletal health of children, in this context, infant allergy and respiratory infections. Fourth, vitamin D level is measured by objective biomarker (maternal or cord serum) in which it reflects the cumulative vitamin D obtained from diet and sun exposure.

Meanwhile, the main limitation of this review was that some outcomes, such as respiratory infections were based on parental reports. This could have caused bias given that self-reports

are not always accurate compared to established tools or medical diagnoses. In addition, all studies involved performed single and not repeated measurements, which may not truly represent the vitamin D levels throughout the entire course of pregnancy. Likewise, all selected studies were derived from Western, high-income and mainly Caucasian populations. Lastly, we did not assess or analyse the heterogeneity of tools and timing across selected studies beyond mere description.

## Conclusion

Overall, evidence on the impacts of vitamin D exposure in utero and in cord blood on the risks of eczema, wheezing and respiratory infections in the first year (12 months) of life is equivocal and inconclusive. This systematic review highlighted that existing data on maternal vitamin D concentrations and eczema, wheezing, and respiratory infections among infants aged 12 months and younger are limited, with no study representing the Asia Pacific region. To form a better understanding and enable a more robust scientific conclusion, more studies using appropriate and comparable methodologies are needed. These include employing randomised control trials or cohort study design with a larger sample size, and measuring the outcomes based on medical diagnoses or established clinical tools to ensure accuracy. In addition, study subjects should comprise a more specific and targeted age group. Measuring vitamin D at multiple period points (instead of single measurement) with appropriate statistical analyses will provide a better understanding of the relationship between the exposure and outcome. Future studies on the role of vitamin D in pregnancy and the early postnatal period in the aetiology of eczema, wheezing and RTI in infants are needed particularly in the Asia Pacific region, as the incidence of infant allergies and respiratory infections is on the rise. Therefore, future intervention study requires a robust study design and appropriate evaluation.

## Supporting information

**S1 Table. This table shows the search terms used for PubMed, MEDLINE, ProQuest, Scopus, CINAHL, the Cochrane Library and Academic Search Premier.**
(PDF)

**S2 Table. This table shows the PRISMA checklist.**
(PDF)

## Author Contributions

**Conceptualization:** Muzaitul Akma Mustapa Kamal Basha, Hazreen Abdul Majid.

**Data curation:** Nuguelis Razali, Abqariyah Yahya.

**Formal analysis:** Muzaitul Akma Mustapa Kamal Basha, Hazreen Abdul Majid.

**Funding acquisition:** Nuguelis Razali.

**Investigation:** Muzaitul Akma Mustapa Kamal Basha.

**Methodology:** Muzaitul Akma Mustapa Kamal Basha, Hazreen Abdul Majid.

**Project administration:** Muzaitul Akma Mustapa Kamal Basha.

**Supervision:** Hazreen Abdul Majid, Abqariyah Yahya.

**Validation:** Hazreen Abdul Majid.

**Visualization:** Abqariyah Yahya.

**Writing – original draft:** Muzaitul Akma Mustapa Kamal Basha.

**Writing – review & editing:** Hazreen Abdul Majid, Nuguelis Razali, Abqariyah Yahya.

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
