## [Decision Letter · Decision Letter 0]

7 Nov 2019

PONE-D-19-24730

Maternal vitamin D status during pregnancy and the risk of eczema, wheezing and respiratory tract infections in the first year of life: a systematic review

PLOS ONE

Dear Dr Abdul Majid,

Thank you for submitting your manuscript to PLOS ONE. After careful consideration, we feel that it has merit but does not fully meet PLOS ONE’s publication criteria as it currently stands. Therefore, we invite you to submit a revised version of the manuscript that addresses the points raised during the review process.

We would appreciate receiving your revised manuscript by Dec 22 2019 11:59PM. To enhance the reproducibility of your results, we recommend that if applicable you deposit your laboratory protocols in protocols.io, where a protocol can be assigned its own identifier (DOI) such that it can be cited independently in the future. For instructions see: http://journals.plos.org/plosone/s/submission-guidelines#loc-laboratory-protocols

We look forward to receiving your revised manuscript.

Kind regards,

Maria Christine Magnus, MPH

Academic Editor

PLOS ONE

Journal Requirements:

2. *Please include in your methods section, the dates included in your search and the dates during which you performed your search.

*Please explain whether you assessed the heterogeneity of the manuscripts.

Additional Editor Comments (if provided):

The reviewer's have raised some important issues that need clarification. I would like to see a revised version of the paper before making a final decision.

Reviewers' comments:

Reviewer's Responses to Questions

**Comments to the Author**

1. Is the manuscript technically sound, and do the data support the conclusions?

Reviewer #1: Partly

Reviewer #2: Yes

Reviewer #3: Yes

2. Has the statistical analysis been performed appropriately and rigorously? 

Reviewer #1: N/A

Reviewer #2: Yes

Reviewer #3: N/A

3. Have the authors made all data underlying the findings in their manuscript fully available?

Reviewer #1: Yes

Reviewer #2: Yes

Reviewer #3: Yes

4. Is the manuscript presented in an intelligible fashion and written in standard English?

Reviewer #1: Yes

Reviewer #2: Yes

Reviewer #3: Yes

5. Review Comments to the Author

Reviewer #1: “Maternal vitamin D status during pregnancy and the risk of eczema, wheezing and respiratory tract infections in the first year of life: a systematic review”

The current study aims to systematically review the literature published until March 2019 on maternal vitamin D status in pregnancy and the development of eczema, wheezing and respiratory tract infections in the offspring within the first year of life. Although the literature search is comprehensive and includes a large number of databases, the rationale for the review in the context of previous studies is unclear, and the presentation of results is not reader friendly. The study also includes studies of cord blood vitamin D, which is not reflected in the title or objectives.

Major comments

The introduction should reflect more strongly that this is a systematic review. Previous reviews and meta-analysis on the current topic need to be summarized and referenced before they appear in the Discussion (refs 26, 43, 44), and there should be a clear description of what the current study contributes. The focus on RTI in the Asia-Pacific region creates some expectations that this review will add evidence from non-western populations, but this does not seem to be the case. It is unclear why the atopic march is brought up, as the current study is limited to infants < =12 months. Greater emphasize could be placed on the potential role of maternal vitamin D status in the etiologies of eczema, wheezing and RTI in the offspring.

Also, the introductory description of the occurrence and time trends in eczema, wheezing and RTI in infants should be more precise with an appropriate choice of references, e.g. references 1 and 2 seem a bit arbitrary (USA) or old (WHO) for documenting the current situation. Judging from the titles, references 3-5 to do not appear to be primary sources of data showing an increase in the rates of over time. General statements such as “…54.7% have been diagnosed with RTIs in early life” is supported by a study from a regional hospital in Cameroon, which is insufficient. The reference list is incomplete with several journal names missing, so revision is needed.

Search strategy and PICOS criteria (Table 1): the objective of the study is to review literature on maternal vitamin D status, but 4 of 10 included studies have assessed vitamin D status in cord blood. The title/scope/Table 1 PICOS statement and study selection criteria could be changed to reflect the inclusion of studies of vitamin D status measured at birth, or these 4 studies should be excluded from the paper. The PICOS statement for “outcome” should include children and age group.

Search terms (Table S1): It seems that the literature search could have been more targeted by including terms for pregnancy, infants or human studies (ref Table S1 PICO). The very broad search may explain why so few studies (10 of 2678 records) were included in the end. It is unclear if search terms were MeSH and/or text words, and why vitamin D deficiency, but not excess (which has also been associated with the outcomes of interest) was included, or chemical forms of vitamin D.

Data extraction (line 131): data on the adjustment for maternal confounders other than parental history should also be extracted and considered when interpreting results and inconsistencies, e.g. were results adjusted for other nutrients of potential importance (e.g. fatty acids or folate) as referenced papers seem to indicate.

Results/study characteristics: it would be helpful if the summary included number of studies vs publications for each outcome, and how these outcomes were assessed (e.g. maternal report or clinical diagnosis) before describing ethnicity of the study populations.

Results in text: reference category needs be mentioned when reporting relative-risk estimates.

Results: Table 2 does not seem space efficient, or reader friendly. Maybe better if split into 2 tables (separate for study characteristics and results where similar outcomes are grouped together)

Results/vitamin D status (line 180-182): a mean value of 14 nmol/L seems unlikely, please verify.

Minor comments:

Results: information regarding exclusion criteria in first paragraph is better suited under Methods.

Results: Some parts of the Table 2 is confusing and may have errors. It seems that LRTI at 15 months is included (Camargo et al 2010) although the study selection criteria is <= 12 months.

Under summary statistics for exposure, what is meant by “(Median 25(OH)D cord blood (IQR) nmol/L = n; 922”?

Abstract: unclear what level of maternal vitamin D that seems to be protective (line 36-37). Also not possible to conclude that evidence is inconsistent due to low number of studies.

Line 189-190: unclear definitions

Line 184: incomplete sentence

Reviewer #2: Overall this is an excellent piece of work and that helps to answer an important question. I think the methodology is in keeping with best practice, and the manuscript is largely well written with appropriate conclusions drawn from the results. The main reason for "major revision" is that I feel there should be assessment and reporting of how outcomes were defined/measured in the the methods and results section. As the authors state in the discussion, this can significantly impact the results and as such should be discussed. If that was adjusted, in addition to some other minor points below, then I would think the manuscript would be ready for publication.

Please see more specific feedback below;

Abstract;

- Clear and well written

Introduction

- The paragraph that begins Page 3, Line 58 is confusing. The paragraph starts with a description of the “atopic march” concept, and then says “similar patterns were observed in RTI cases”. However RTI cases in early life do not lead to more severe persistent RTI cases in later life, so I do not think there is a similar pattern to atopic march.

I think what the authors are trying to say is that atopic disease and RTI are both common in early life, and important causes of morbidity as atopic disease can “march” on to more severe, persistent atopic disease and RTI is a leading cause of hospital admission. If so, I agree with that, but I think the paragraph needs reworking.

- This is a minor point, but the final paragraph of the introduction would have more of an impact of it consisted of the 2 sentences. The one the starts “to this day …” (page 4, line 82) and “this systematic review aimed …” (page 4 line 85). The points re importance of measuring serum vitamin D rather than dietary is important and valid, but could be worked into the previous paragraph.

Methods

- well done

- clear question, followed best practice, registered prospectively

- Please reference the table with the search terms in the data sources paragraph so that readers know it is available.

- assessment of definition of each outcome is important as this is a potential source of heterogeneity (i.e was eczema /wheezing/RTI parent report, doctor diagnosed, or assessed by the trial team)

Results

- Please reference the PRISMA diagram at the start of the results section, so that readers know that is available for them to look at. Same for PRISMA checklist

- need to incorporate comment on how each outcome was assessed. See feedback above

Discussion

- minor point; page 22, line 261 “- when put together-“ would be better to have “,” rather than “-“ (i.e , when put together,)

- page 23, line 280; It would be good to briefly outline what the results of prior systematic reviews are, that this review contradicts

- page 23, line 283; matured should be mature

- page 23, line 284; there is a full stop missing before “Second”

- page 23, line 285; this is an important point (method of outcome assessment) but should not be mentioned for the first time here. It should be incorporated into both methods and results.

Conclusion

- Page 24; Line 314; Include examples of what more appropriate and comparable methodologies would include (i.e multiple measurements of vitamin D, with statistical analysis involving … and outcomes assessed …)

Reviewer #3: This is a well written perspective that discusses the evidence for maternal vitamin D levels controlling the incidence of eczema, wheezing and respiratory tract infections in infants before one year of life. The maternal levels were measured at times ranging from during the first trimester to ‘at birth’ with measures in the cord blood. Vitamin D levels from a single sample were sometimes published as absolute amounts but in other studies only defined into categories. Another limitation was that some outcomes were based on parental reports and were not necessarily nurse or doctor defined. Ten studies were scrutinised in full, all of which included primarily western/Caucasian populations. Measures of maternal vitamin D were considered as a possible determinant of infant eczema, wheezing or respiratory tract infections; the discussion for this study did not consider vitamin D levels of the infant per se as a possible contributor to allergic outcomes. Perhaps vitamin D deficiency during the first year of life is as important as the level before birth.

The review is informative and suggests that no robust association between maternal vitamin D and infant allergy exists. If an effect exists, it is minor and can be detected only inconsistently in studies. It will be interesting if the authors perform a similar analysis in the Malaysian population; such a study would be very informative due to the latitude of Malaysia and the Malaysians recruited to the study will have obtained their vitamin D levels from significant sun exposure. Many of the other studies reported included populations living at higher latitudes. As inferred in line 89, there is a growing awareness that vitamin D levels are a biomarker of recent sun exposure rather than being the active molecule in the condition under study. Any association of vitamin D levels with a condition cannot infer that vitamin D is the causal molecule. Instead, other sun-induced molecules may be responsible for the benefit found. In a revised manuscript, this argument should be included in the Discussion.

Line 151, remove with.

Line 184, something is missing, Three what?

Line 280, references are required for the prior systematic reviews.

6. PLOS authors have the option to publish the peer review history of their article (what does this mean?). If published, this will include your full peer review and any attached files.

Reviewer #1: No

Reviewer #2: Yes: Shivanthan Shanthikumar

Reviewer #3: No

---

## [Author Response · Author response to Decision Letter 0]

19 Dec 2019

“Author Response”

We would like to express our sincere gratitude to the esteemed reviewers for their useful comments. We have now revised the manuscript according to the reviewers’ suggestions, and sincerely hope that the manuscript is now suitable for publication in Plos One.

Editor Comments

 Thank you for highlighting this. We had amended it accordingly. 

2. *Please include in your methods section, the dates included in your search and the dates during which you performed your search.

 The information that is requested already been updated in method section; Line 104-106

 “The following databases were used to search for studies published until March 31, 2019; PubMed, MEDLINE, 

 ProQuest, Scopus, CINAHL, the Cochrane Library and Academic Search Premier. This search was performed on May 1, 

 2019.” 

3. *Please explain whether you assessed the heterogeneity of the manuscripts. 

 Thank you for pointing that out. We did not use any scoring to assess heterogeneity. However, from the definition of 

 outcomes evaluation, assessment for heterogeneity is by evaluating the 10 papers ( 8 studies reported the 

 measurement of the outcomes were by parental reported and only 2 reported using established scoring tool. Hence 

 because of that, we updated as limitation. 

Comments from Reviewer 1

1. The current study aims to systematically review the literature published until March 2019 on maternal vitamin D status in pregnancy and the development of eczema, wheezing and respiratory tract infections in the offspring within the first year of life. Although the literature search is comprehensive and includes a large number of databases, the rationale for the review in the context of previous studies is unclear, and the presentation of results is not reader friendly. The study also includes studies of cord blood vitamin D, which is not reflected in the title or objectives.

 Thank you for pointing that out. We appreciate the reviewer comments and we have revised some sentences, words, 

 title and objectives of the review. We have amended the title; “Risk of eczema, wheezing and respiratory tract 

 infections in the first year of life: a systematic review of vitamin D concentrations during pregnancy and at birth”

 We also have updated the amendment and revision to PROSPERO with registration no. CRD42018093039. 

2. The introduction should reflect more strongly that this is a systematic review. 

 We acknowledged your input and the introduction had been revised accordingly to show this is a systematic review.

3. Previous reviews and meta-analysis on the current topic need to be summarized and referenced before they appear in 

 the Discussion (refs 26, 43, 44), and there should be a clear description of what the current study contributes. 

 Thank you for pointing that out. The information have been added accordingly in the introductory section; Line 71- 

 75. “To this day, it is still unclear to what extent the vitamin D levels during pregnancy or early postnatal (e.g, cord 

 blood) affects the development of allergic diseases and RTIs in infants. Prior reviews have included a wide range of 

 age while assessing allergic diseases and RTI outcomes (e.g 0 to 5 years old), and employed different methods in 

 vitamin D measurement; serum 25(OH)D and dietary intake (4, 23, 26).

4.The focus on RTI in the Asia-Pacific region creates some expectations that this review will add evidence from non-western populations, but this does not seem to be the case. 

 Thank you for highlighting this. It would be a good input if studies had been conducted in Asia Pacific region. We did 

 not found any. A study conducted in Japan had to be excluded as had met our exclusion criteria.

5. It is unclear why the atopic march is brought up, as the current study is limited to infants < =12 months. 

 Thank you for pointing that out. We take note it and we had removed it.

6. Greater emphasize could be placed on the potential role of maternal vitamin D status in the etiologies of eczema, wheezing and RTI in the offspring.

 Thank you for highlighting this. Noted that vitamin D has potential role in bone’s health, other musculoskeletal health 

 such as rickets, cardiovascular disease, diabetes and certain cancers. However it is not strong enough to show the 

 association of high vitamin levels and the eczema status. Thus we could not put a greater emphasize literature on this 

 issue. 

7. Also, the introductory description of the occurrence and time trends in eczema, wheezing and RTI in infants should be more precise with an appropriate choice of references, e.g. references 1 and 2 seem a bit arbitrary (USA) or old (WHO) for documenting the current situation. Judging from the titles, references 3-5 to do not appear to be primary sources of data showing an increase in the rates of over time. General statements such as “…54.7% have been diagnosed with RTIs in early life” is supported by a study from a regional hospital in Cameroon, which is insufficient. 

 Thank you for highlighting this. We had removed that points and we had amended it. 

8. The reference list is incomplete with several journal names missing, so revision is needed.

 Thank you for pointing that out. We had revised and had corrected it.

9. Search strategy and PICOS criteria (Table 1): the objective of the study is to review literature on maternal vitamin D status, but 4 of 10 included studies have assessed vitamin D status in cord blood. The title/scope/Table 1 PICOS statement and study selection criteria could be changed to reflect the inclusion of studies of vitamin D status measured at birth, or these 4 studies should be excluded from the paper. The PICOS statement for “outcome” should include children and age group.

 Thank you for the reviewer comment and suggestion. We had revised it and had made necessary changes. We have 

 amendment the title and had been changed accordingly. The title/scope/Table 1 PICOS statement and study selection 

 criteria had been updated and had amended in PROSPERO.

10. Search terms (Table S1): It seems that the literature search could have been more targeted by including terms for pregnancy, infants or human studies (ref Table S1 PICO). The very broad search may explain why so few studies (10 of 2678 records) were included in the end. 

 Thank you for the suggestion. We used a larger search term to ensure we did not miss the evidence by specifically. 

 Having specific search term, we afraid it may reduce paper that will be selected. 

11. It is unclear if search terms were MeSH and/or text words, and why vitamin D deficiency, but not excess (which has also been associated with the outcomes of interest) was included, or chemical forms of vitamin D.

 Thank you for pointing that out. We used MESH term and this had already explicitly in line 109-111. 

12. Data extraction (line 131): data on the adjustment for maternal cofounders other than parental history should also be extracted and considered when interpreting results and inconsistencies, e.g. were results adjusted for other nutrients of potential importance (e.g. fatty acids or folate) as referenced papers seem to indicate.

 Thank you for highlighting this. we had added in the table result accordingly.

13. Results/study characteristics: it would be helpful if the summary included number of studies vs publications for each outcome, and how these outcomes were assessed (e.g. maternal report or clinical diagnosis) before describing ethnicity of the study populations.

 Thank you for highlighting this. The information that is required already been updated. “Methods of outcome 

 assessment were diverse. Most studies relied on parental reports (n=8 studies) while some used standardized tools 

 (n=2 studies). The two tools used to determine eczema were the ‘SCORing Atopic Dermatitis’ (SCORAD) index (7) 

 and the modified version of the UK Working Party’s diagnostic criteria (35).

14. Results in text: reference category needs be mentioned when reporting relative-risk estimates.

 Thank you for highlighting this. The information that is required already been updated. 

15. Results: Table 2 does not seem space efficient, or reader friendly. Maybe better if split into 2 tables (separate for study characteristics and results where similar outcomes are grouped together)

 Thank you for highlighting this. We appreciate the reviewer suggestion and we have simplified the table accordingly.

15. Results/vitamin D status (line 180-182): a mean value of 14 nmol/L seems unlikely, please verify.

 Thank you for pointing that out. We take note it and had revised and made necessary changes. Line 186-187. 

 “The average concentration of serum 25(OH)D during pregnancy ranged 50.0 to 73.6 nmol/L and early postnatal (in 

 cord blood) ranged from 44.0 to 58.4 nmol/L.”

16. Results: information regarding exclusion criteria in first paragraph is better suited under Methods.

 Thank you for the reviewer comment. The information has been removed and changes has been made accordingly in 

 Method section.

17. Results: Some parts of the Table 2 is confusing and may have errors. It seems that LRTI at 15 months is included (Camargo et al 2010) although the study selection criteria is <= 12 months.

 Thank you for pointing that out. Thank you for the reviewer comment. The information had been removed and 

 changes has been made accordingly.

18. Under summary statistics for exposure, what is meant by “(Median 25(OH)D cord blood (IQR) nmol/L = n; 922”?

 Thank you for pointing that out. The information had been added.

19. Abstract: unclear what level of maternal vitamin D that seems to be protective (line 36-37). Also not possible to conclude that evidence is inconsistent due to low number of studies.

 Thank you for pointing that out. We had revised and had made necessary changes accordingly. Line 39-42

 “Studies on the associations between vitamin D concentrations during pregnancy and at birth and the mentioned 

 outcomes among infants were found limited. Nevertheless, there is conflicting evidence showing the risk of eczema, 

 wheezing and RTIs with vitamin D concentration in early life.”

20. Line 189-190: unclear definitions 

 Thank you for pointing that out. We have gone through and make necessary changes. Line 193 – 196.

 “In determining the associations between exposure (vitamin D concentrations during pregnancy or early postnatal) 

 and outcomes (allergic eczema, wheezing or RTIs), some studies used quartiles and quintile while some employed 

 cut-off values with different presentation of vitamin D concentration categorizations, such as deficient (25.0-40.0, < 

 50.0 nmol/L),…”

21. Line 184: incomplete sentence 

 Thank you for pointing that out. The information had been added. Line 190-191. “Three studies reported mean and 

 standard deviation for the serum 25(OH)D results…”

Comments from Reviewer 2

1. Overall this is an excellent piece of work and that helps to answer an important question. I think the methodology is in keeping with best practice, and the manuscript is largely well written with appropriate conclusions drawn from the results. 

The main reason for "major revision" is that I feel there should be assessment and reporting of how outcomes were defined/measured in the methods and results section. As the authors state in the discussion, this can significantly impact the results and as such should be discussed. If that was adjusted, in addition to some other minor points below, then I would think the manuscript would be ready for publication.

 Thank you for pointing that out. We appreciate the reviewer suggestion and we have corrected and make necessary 

 changes. The outcomes were defined as in Line 162 – 165.

2. Abstract - Clear and well written

 Thank you for the comment.

3. Introduction:

- The paragraph that begins Page 3, Line 58 is confusing. The paragraph starts with a description of the “atopic march” concept, and then says “similar patterns were observed in RTI cases”. However RTI cases in early life do not lead to more severe persistent RTI cases in later life, so I do not think there is a similar pattern to atopic march. I think what the authors are trying to say is that atopic disease and RTI are both common in early life, and important causes of morbidity as atopic disease can “march” on to more severe, persistent atopic disease and RTI is a leading cause of hospital admission. If so, I agree with that, but I think the paragraph needs reworking. This is a minor point, but the final paragraph of the introduction would have more of an impact of it consisted of the 2 sentences. The one the starts “to this day …” (page 4, line 82) and “this systematic review aimed …” (page 4 line 85). 

 Thank you for pointing that out. We have gone through and make necessary changes. We have removed the 

 information of “atopic march” paragraph. This has been amended and comment were incorporated in introduction 

 section (Line 58-62).

The points re importance of measuring serum vitamin D rather than dietary is important and valid, but could be worked into the previous paragraph.

 Thank you for highlighting this. We had revised and had amended accordingly. Line 76- 79.

 “Clinically, serum 25hydroxyvitamin D (25(OH)D) is used as a marker of vitamin D level because it represents the 

 cumulative effect of dietary intake of vitamin D and sunlight exposure (27). Thus, dietary intake alone might not be a 

 good indicator of overall vitamin D levels as ultraviolet B-rays induces skin formation of vitamin D.”

4. Methods

- well done

- clear question, followed best practice, registered prospectively

- Please reference the table with the search terms in the data sources paragraph so that readers know it is available.

- assessment of definition of each outcome is important as this is a potential source of heterogeneity (i.e was eczema /wheezing/RTI parent report, doctor diagnosed, or assessed by the trial team) 

Thank you for the comment. We take note it and had referenced the table with the search terms as in line 111.

The description on how outcomes were assessed by the studies had been updated as in line 162– 165.

5. Results

- Please reference the PRISMA diagram at the start of the results section, so that readers know that is available for them to look at. Same for PRISMA checklist.

- Need to incorporate comment on how each outcome was assessed. See feedback above

 Thank you for pointing that out.

 We take note it and had referenced the PRISMA diagram at the start of the results section as stated in line 155 – 

 Figure 1. Figure 1 is the PRISMA diagram.

6. Discussion

- minor point; page 22, line 261 “- when put together-“ would be better to have “,” rather than “-“ (i.e , when put together,)

 Thank you for pointing that out and the suggestion. We have corrected it. Line 294.

- page 23, line 280; It would be good to briefly outline what the results of prior systematic reviews are, that this review contradicts

 Thank you for pointing that out. We have gone through and make necessary changes. Line 308-310.

 “Previous reviews indicated a preventive role of vitamin D in early life against wheezing and RTIs; inverse 

 associations were found between vitamin D levels in early life and wheezing and/or RTIs among infants (23, 26).”

- page 23, line 283; matured should be mature 

 Thank you for pointing that out and the suggestion. We have corrected it.

- page 23, line 284; there is a full stop missing before “Second” 

 Thank you for pointing that out and the suggestion. We have corrected it. 

- page 23, line 285; this is an important point (method of outcome assessment) but should not be mentioned for the first time here. It should be incorporated into both methods and results.

 Thank you for pointing that out. We have gone through and make necessary changes. We added information of method of the outcome assessment in Result/study characteristic section. Line 163 – 171.

7. Conclusion

- Page 24; Line 314; Include examples of what more appropriate and comparable methodologies would include (i.e multiple measurements of vitamin D, with statistical analysis involving … and outcomes assessed …)

 Thank you for pointing that out. We have gone through and added a few points. Line 354-359.

 “These include employing randomized control trial or cohort study design with a larger sample size, and measuring 

 the outcomes based on medical diagnosis or established clinical tools to ensure its’ accuracy. In addition, the study 

 subjects should comprise a more specific and targeted age group….”

Comments from Reviewer 3

1. This is a well written perspective that discusses the evidence for maternal vitamin D levels controlling the incidence of eczema, wheezing and respiratory tract infections in infants before one year of life. The maternal levels were measured at times ranging from during the first trimester to ‘at birth’ with measures in the cord blood. Vitamin D levels from a single sample were sometimes published as absolute amounts but in other studies only defined into categories.

 Noted with thanks.

2. Another limitation was that some outcomes were based on parental reports and were not necessarily nurse or doctor defined. 

 I agree with your comment and this is highlighted in our limitation section.

3. Ten studies were scrutinised in full, all of which included primarily western/Caucasian populations. Measures of maternal vitamin D were considered as a possible determinant of infant eczema, wheezing or respiratory tract infections; the discussion for this study did not consider vitamin D levels of the infant per se as a possible contributor to allergic outcomes. Perhaps vitamin D deficiency during the first year of life is as important as the level before birth.

The review is informative and suggests that no robust association between maternal vitamin D and infant allergy exists. If an effect exists, it is minor and can be detected only inconsistently in studies. It will be interesting if the authors perform a similar analysis in the Malaysian population; such a study would be very informative due to the latitude of Malaysia and the Malaysians recruited to the study will have obtained their vitamin D levels from significant sun exposure. 

 Your constructive suggestions is noted and the study among infant in Malaysia will be conducted in near future. 

4. Many of the other studies reported included populations living at higher latitudes. As inferred in line 89, there is a 

growing awareness that vitamin D levels are a biomarker of recent sun exposure rather than being the active 

molecule in the condition under study. Any association of vitamin D levels with a condition cannot infer that vitamin D 

is the causal molecule. Instead, other sun-induced molecules may be responsible for the benefit found. In a revised manuscript, this argument should be included in the Discussion.

 Thank you for the suggestions. I agreed that potentially there are other molecule that is activated by sun that may 

 confer benefits. However the literature on this is scarce. Nevertheless, despite some countries at the equator and 

 have sun exposure most of time, the population/ the subject can also be at risk of vitamin D insufficiency due to the 

 clothing style as reported by recent study, Quah et al. (2018).

5. Line 151, remove with. 

 Thank you for pointing that out. We have removed “with”.

6. Line 184, something is missing, Three what? 

 Thank you for pointing that out. We have added the missing sentence.

7. Line 280, references are required for the prior systematic reviews. 

Thank you for pointing that out. We take note it, and had added the references.

---

## [Decision Letter · Decision Letter 1]

12 Feb 2020

PONE-D-19-24730R1

Risk of eczema, wheezing and respiratory tract infections in the first year of life: a systematic review of vitamin D concentrations during pregnancy and at birth

PLOS ONE

Dear Dr Majid,

Thank you for submitting your manuscript to PLOS ONE. After careful consideration, we feel that it has merit but does not fully meet PLOS ONE’s publication criteria as it currently stands. Therefore, we invite you to submit a revised version of the manuscript that addresses the points raised during the review process.

We would appreciate receiving your revised manuscript by Mar 28 2020 11:59PM. To enhance the reproducibility of your results, we recommend that if applicable you deposit your laboratory protocols in protocols.io, where a protocol can be assigned its own identifier (DOI) such that it can be cited independently in the future. For instructions see: http://journals.plos.org/plosone/s/submission-guidelines#loc-laboratory-protocols

We look forward to receiving your revised manuscript.

Kind regards,

Maria Christine Magnus, MPH

Academic Editor

PLOS ONE

Additional Editor Comments (if provided):

As stated by two of the reviewers, the manuscript needs substantial editing of the language. Please consider having a native english speaker read through the manuscript.

Reviewers' comments:

Reviewer's Responses to Questions

**Comments to the Author**

1. If the authors have adequately addressed your comments raised in a previous round of review and you feel that this manuscript is now acceptable for publication, you may indicate that here to bypass the “Comments to the Author” section, enter your conflict of interest statement in the “Confidential to Editor” section, and submit your "Accept" recommendation.

Reviewer #1: (No Response)

Reviewer #2: All comments have been addressed

Reviewer #3: (No Response)

2. Is the manuscript technically sound, and do the data support the conclusions?

Reviewer #1: Yes

Reviewer #2: Yes

Reviewer #3: Partly

3. Has the statistical analysis been performed appropriately and rigorously? 

Reviewer #1: N/A

Reviewer #2: Yes

Reviewer #3: Yes

4. Have the authors made all data underlying the findings in their manuscript fully available?

Reviewer #1: Yes

Reviewer #2: (No Response)

Reviewer #3: Yes

5. Is the manuscript presented in an intelligible fashion and written in standard English?

Reviewer #1: No

Reviewer #2: Yes

Reviewer #3: No

6. Review Comments to the Author

Reviewer #1: Comments to paper after first revision:

Although the methodology for the literature search seems solid, the revised manuscript and presentation of results seem to suffer from typos, errors and inconsistencies and leaves an untidy impression. Also, it still remains unclear what the current study adds compared with the previous reviews and meta-analyses referenced (refs 4, 23, 26); e.g. does the current study include more studies on wheeze in infants than the meta-analysis from 2018 (ref 4) or do you summarize the evidence differently?

The authors argue that previous reviews have included vitamin D intake, but this is not a problem as long as vitamin D intake and vitamin D status are presented as separate analyzes, which is usually the case –without knowing these studies in detail. Regarding the age restriction of all outcomes (RTI, eczema, wheeze) to infants, the rationale for doing this should be made more clear, especially because the authors conclude that the literature is limited and the outcomes of eczema and wheeze also occur in children after the first year of life. Some arguments now mentioned in the Discussion, could be moved to the Introduction. However, for vitamin D there is still a chance that prenatal exposure may play a role, due to vitamin D supplement recommendations to infants in some countries, and the introduction of complementary feeding which may be enriched with vit D.

Major comments/

The study rationale could be refined some more.

The manuscript needs proof reading from the very beginning, a few examples include the Abstract (line 29 – duplicate words, missing commas; line 36 missing “in” or “during”) and Introduction (line 50, grammar). Clarity is affected in some places, e.g. in the Introduction (line 57) – it is unclear if “allergy eczema” should read “allergy and eczema” or “allergic eczema”. There are now 2 tables labelled “Table 3”, including the second page of Table 2, which should be corrected.

The paper is currently missing some information on outcome definitions as part of the inclusion/exclusion criteria. What types of eczema, wheeze, and RTI were included in the analysis and how were the outcomes assessed? Results on eczema (Table2/Table 3) seem to be presented for both eczema and atopic eczema, and RTI (Table 5) seems to include several types, including LRTI (pneumonia/bronchiolitis), URTI, recurrent RTI, as well as symptoms of cough. Now the information is briefly mentioned in the Discussion (line 286, 287). Unclear why ”Pregnancy outcomes” are mentioned in Table 2/3 (ref 38).

Tables have improved, but the information seems to be provided in an inconsistent manner. If the information was not available, it should be explained somewhere. The conclusion should preferably reflect the conclusions of the authors of the review (not the conclusions found in the papers), and be reported in a similar manner for all papers to the extent possible.

Taking Table 3 as an example:

Gale et al: no sample size provided, sig results for Q4 not marked in bold (but some results marked in bold in Table 4)

Weisse et al: no sample size provided, unclear if adj ORs refer to highest vs. lowest category

Gazibara et al. unclear if “n” refers to no. of cases or study size

Baiz et al. sample size provided twice (n=239), unclear if the reported OR is per nmol/L of vitamin D or if exposure categories are missing

Summary of results: it is hard to follow at times. The authors could consider summarizing the number of studies reporting no/positive/negative associations for each outcome.

Discussion/Vit D & RTI: although sex of the child may affect the risk of RTI, sex is not expected to be associated with maternal vit D status or vit D status at birth, and therefore not a confounder by the usual definition. Thus, lack of adjustment for sex should not be considered a weakness of studies.

Minor comments

Reference no. 9 to document prevalence (lines 52, 56), is based on mathematical modelling and not original data, which should be stated explicitly

Table 4: could go in the supplement with Table S3

Tables 3-5 titles: Suggest changing for better clarity from “Studies included in the analysis between 25 (OH)D during pregnancy and cord blood and the development of … in the first year (12 months) of life” to “Studies included in the analysis of 25 (OH)D during pregnancy or cord blood with the development of … in the first year (12 months) of life”.

Discussion/Vitamin D status: unclear what is meant by “evidence has shown that vitamin D concentrations are a biomarker of sun exposure” because you also write that vitamin D concentrations reflect the “cumulative effect of vitamin D dietary intake and sun

exposure”

Reviewer #2: Thanks you for the revisions. I thin you addressed most of my concerns. There are a few minor points;

1. Page 10 - heading of the table is "Table 3 Characteristics of 10 studies included in the analysis between 25 (OH)D during pregnancy and cord blood and the development of eczema, wheezing and RTIs in the first year (12 months) of life (continue)" - should this be table 2 (not table 3)?

2. I think it would be good to include a column in Table 2 on how the each outcome was assessed, even if it is just health professional diagnosed or parent reported

3. I think the results section, under the headings Vitamin D and Eczema in the first year (12 months) of life, Vitamin D and wheezing in the first year (12 months) of life, and Vitamin D and respiratory infections in the first year (12 months) of life it would be good to comment on whether the method of outcome assessment could explain the conflicting results found.

Reviewer #3: This is an improved manuscript but the English spelling and phrases are very poor. Much of the manuscript requires rewriting. Many of the corrections required are listed below but perhaps an English writer should be consulted.

Line 36: During should be removed.

Line 50: continued to increase.

Introduction: 3rd sentence is repetitive of the 2nd sentence.

Line 56: years 2014 to 2015.

Line 57: ranged from xx

Line 62: The majority.

Line 64: second para, reword the sentence.

Line 66: ‘low’ vitamin D status

Line 68: remove ‘the’

Line 72: Cord blood does not represent early postnatal. In many places (for example line 107), cord blood is clearly stated as at birth, rather than early postnatal.

Line 81: remove course

Line 87: remove pool

Line 88: Low vitamin D levels are the risk factor

Line 129: ‘the’

Line 187: ranged ‘from’

Line 188: the meaning of ‘timing of exposure assessment’ is not clear

Line 210: Meaning of late of pregnancy?

Line 263: remove ‘There are’

Line 268: It is associations between X and Y

Lines 278-280: rewrite as meaning not clear

Line 282: populations

Lines 287-288: The literature is not scarce for the properties of other molecules induced by UV radiation. See Hart et al, Nat Rev Immunol 2011, and Hart et al, Annu Rev Pathol 2019.

Lines 288-290: not good English, please rewrite.

7. PLOS authors have the option to publish the peer review history of their article (what does this mean?). If published, this will include your full peer review and any attached files.

Reviewer #1: No

Reviewer #2: Yes: Shivanthan ShanthikumarDr

Reviewer #3: No

---

## [Author Response · Author response to Decision Letter 1]

22 Mar 2020

Reviewer #1: Comments to paper after first revision:

1. Although the methodology for the literature search seems solid, the revised manuscript and presentation of results seem to suffer from typos, errors and inconsistencies and leaves an untidy impression. 

Author Response: Thank you for highlighting this. We had amended it accordingly.

2. Also, it still remains unclear what the current study adds compared with the previous reviews and meta-analyses referenced (refs 4, 23, 26); e.g. does the current study include more studies on wheeze in infants than the meta-analysis from 2018 (ref 4) or do you summarize the evidence differently?

Author Response: The information that is requested already been updated in method section; Line 80-98

3. The authors argue that previous reviews have included vitamin D intake, but this is not a problem as long as vitamin D intake and vitamin D status are presented as separate analyzes, which is usually the case –without knowing these studies in detail. Regarding the age restriction of all outcomes (RTI, eczema, wheeze) to infants, the rationale for doing this should be made more clear, especially because the authors conclude that the literature is limited and the outcomes of eczema and wheeze also occur in children after the first year of life. Some arguments now mentioned in the Discussion, could be moved to the Introduction. However, for vitamin D there is still a chance that prenatal exposure may play a role, due to vitamin D supplement recommendations to infants in some countries, and the introduction of complementary feeding which may be enriched with vit D.

Author Response: The information that is requested already been updated in method section; Line 80-84; 92-95. This is further elaborated to highlight the importance of current literature review.

Major comments:

4. The study rationale could be refined some more.

Author Response: The information that is requested already been updated in method section; Line 80-84; 92-98.

5. The manuscript needs proof reading from the very beginning, a few examples include the Abstract (line 29 – duplicate words, missing commas; line 36 missing “in” or “during”) and Introduction (line 50, grammar). Clarity is affected in some places, e.g. in the Introduction (line 57) – it is unclear if “allergy eczema” should read “allergy and eczema” or “allergic eczema”. There are now 2 tables labelled “Table 3”, including the second page of Table 2, which should be corrected.

Author Response: Thank you for highlighting this. The manuscript was proofread. We had amended and corrected it accordingly.

6. The paper is currently missing some information on outcome definitions as part of the inclusion/exclusion criteria. What types of eczema, wheeze, and RTI were included in the analysis and how were the outcomes assessed? Results on eczema (Table2/Table 3) seem to be presented for both eczema and atopic eczema, and RTI (Table 5) seems to include several types, including LRTI (pneumonia/bronchiolitis), URTI, recurrent RTI, as well as symptoms of cough. Now the information is briefly mentioned in the Discussion (line 286, 287). 

Author Response: The information that is requested already been updated in method section; Line 128-131; 148-151

7. Unclear why ”Pregnancy outcomes” are mentioned in Table 2/3 (ref 38).

Author Response: Thank you for highlighting this. We had removed it.

8. Tables have improved, but the information seems to be provided in an inconsistent manner. If the information was not available, it should be explained somewhere. The conclusion should preferably reflect the conclusions of the authors of the review (not the conclusions found in the papers), and be reported in a similar manner for all papers to the extent possible.

Author Response: Thank you for highlighting this. We had amended it accordingly.

9. Taking Table 3 as an example:

Gale et al: no sample size provided, sig results for Q4 not marked in bold (but some results marked in bold in Table 4)

Author Response: Thank you for highlighting this. We had corrected and bold it.

10. Weisse et al: no sample size provided, unclear if adj ORs refer to highest vs. lowest category

Author Response: The information already been updated.

11. Gazibara et al. unclear if “n” refers to no. of cases or study size

Author Response: Thank you for highlighting this. We had amended it. We also added a footnotes. N=study size; n=number of cases.

12. Baiz et al. sample size provided twice (n=239), unclear if the reported OR is per nmol/L of vitamin D or if exposure categories are missing

Author Response: We had removed it and the information been updated in the table.

13. Summary of results: it is hard to follow at times. The authors could consider summarizing the number of studies reporting no/positive/negative associations for each outcome.

Author Response: Thank you for the suggestions. We had amended it accordingly.

14. Discussion/Vit D & RTI: although sex of the child may affect the risk of RTI, sex is not expected to be associated with maternal vit D status or vit D status at birth, and therefore not a confounder by the usual definition. Thus, lack of adjustment for sex should not be considered a weakness of studies.

Author Response: Thank you for highlighting this. We had removed and amended it accordingly. Line 315-321.

Minor comments

15. Reference no. 9 to document prevalence (lines 52, 56), is based on mathematical modelling and not original data, which should be stated explicitly

Author Response: Thank you for highlighting this. We had amended it accordingly. Line 56-59.

16. Table 4: could go in the supplement with Table S3

Author Response: Thank you for the suggestion. 

17. Tables 3-5 titles: Suggest changing for better clarity from “Studies included in the analysis between 25 (OH)D during pregnancy and cord blood and the development of … in the first year (12 months) of life” to “Studies included in the analysis of 25 (OH)D during pregnancy or cord blood with the development of … in the first year (12 months) of life”.

Author Response: Thank you for the suggestion. We had amended it accordingly.

18. Discussion/Vitamin D status: unclear what is meant by “evidence has shown that vitamin D concentrations are a biomarker of sun exposure” because you also write that vitamin D concentrations reflect the “cumulative effect of vitamin D dietary intake and sun exposure”.

Author Response: Thank you for highlighting this. We had removed the sentence and the sentences were rephrased accordingly. Line 273-282.

Reviewer #2: 

Thanks you for the revisions. I thin you addressed most of my concerns. There are a few minor points;

1. Page 10 - heading of the table is "Table 3 Characteristics of 10 studies included in the analysis between 25 (OH)D during pregnancy and cord blood and the development of eczema, wheezing and RTIs in the first year (12 months) of life (continue)" - should this be table 2 (not table 3)?

Author Response: Thank you for highlighting this. We had renumbered it accordingly. 

2. I think it would be good to include a column in Table 2 on how the each outcome was assessed, even if it is just health professional diagnosed or parent reported

Author Response: Thank you for the suggestion. We had added it accordingly in Table 2.

3. I think the results section, under the headings Vitamin D and Eczema in the first year (12 months) of life, Vitamin D and wheezing in the first year (12 months) of life, and Vitamin D and respiratory infections in the first year (12 months) of life it would be good to comment on whether the method of outcome assessment could explain the conflicting results found.

Author Response: Thank you for the suggestion. We had added the information accordingly. Line 220-222; 226-227.

Reviewer #3: 

This is an improved manuscript but the English spelling and phrases are very poor. Much of the manuscript requires rewriting. Many of the corrections required are listed below but perhaps an English writer should be consulted.

Line 36: During should be removed.

Line 50: continued to increase.

Introduction: 3rd sentence is repetitive of the 2nd sentence.

Line 56: years 2014 to 2015.

Line 57: ranged from xx

Line 62: The majority.

Line 64: second para, reword the sentence.

Line 66: ‘low’ vitamin D status

Line 68: remove ‘the’

Line 72: Cord blood does not represent early postnatal. In many places (for example line 107), cord blood is clearly stated as at birth, rather than early postnatal.

Line 81: remove course

Line 87: remove pool

Line 88: Low vitamin D levels are the risk factor

Line 129: ‘the’

Line 187: ranged ‘from’

Line 188: the meaning of ‘timing of exposure assessment’ is not clear

Line 210: Meaning of late of pregnancy?

Line 263: remove ‘There are’

Line 268: It is associations between X and Y

Lines 278-280: rewrite as meaning not clear

Line 282: populations

Lines 287-288: The literature is not scarce for the properties of other molecules induced by UV radiation. See Hart et al, Nat Rev Immunol 2011, and Hart et al, Annu Rev Pathol 2019.

Lines 288-290: not good English, please rewrite.

Author Response: Thank you for highlighting this. Noted with thank you. The manuscript was proofread. We had amended and corrected it accordingly.

---

## [Decision Letter · Decision Letter 2]

17 Apr 2020

PONE-D-19-24730R2

Risk of eczema, wheezing and respiratory tract infections in the first year of life: a systematic review of vitamin D concentrations during pregnancy and at birth

PLOS ONE

Dear Dr Majid,

Thank you for submitting your manuscript to PLOS ONE. After careful consideration, we feel that it has merit but does not fully meet PLOS ONE’s publication criteria as it currently stands. Therefore, we invite you to submit a revised version of the manuscript that addresses the points raised during the review process.

We would appreciate receiving your revised manuscript by Jun 01 2020 11:59PM. To enhance the reproducibility of your results, we recommend that if applicable you deposit your laboratory protocols in protocols.io, where a protocol can be assigned its own identifier (DOI) such that it can be cited independently in the future. For instructions see: http://journals.plos.org/plosone/s/submission-guidelines#loc-laboratory-protocols

We look forward to receiving your revised manuscript.

Kind regards,

Maria Christine Magnus, MPH

Academic Editor

PLOS ONE

Reviewers' comments:

Reviewer's Responses to Questions

**Comments to the Author**

1. If the authors have adequately addressed your comments raised in a previous round of review and you feel that this manuscript is now acceptable for publication, you may indicate that here to bypass the “Comments to the Author” section, enter your conflict of interest statement in the “Confidential to Editor” section, and submit your "Accept" recommendation.

Reviewer #1: (No Response)

Reviewer #3: (No Response)

2. Is the manuscript technically sound, and do the data support the conclusions?

Reviewer #1: Yes

Reviewer #3: Yes

3. Has the statistical analysis been performed appropriately and rigorously? 

Reviewer #1: N/A

Reviewer #3: Yes

4. Have the authors made all data underlying the findings in their manuscript fully available?

Reviewer #1: Yes

Reviewer #3: Yes

5. Is the manuscript presented in an intelligible fashion and written in standard English?

Reviewer #1: No

Reviewer #3: Yes

6. Review Comments to the Author

Reviewer #1: Comments to paper after second revision.

The authors have done a thorough revision. The study rationale is much now clearer and the manuscript reads a lot better. Most of my comments have been addressed and tables have improved considerably. A few minor points remain regarding the reporting of the results.

Table 2: no apparent order of studies. Could be sorted by publication year.

The marking of significant ORs in bold does still not seem entirely consistent, e.g. in Table 4 significant ORs are unmarked for the studies by De Jongh (OR for Q2) and Baiz. In Table 5 the same applies to the studies by De Jongh (Q3) and by Camargo.

It remains unclear for some studies what unit/increase the ORs are reported for, e.g. Table 4 for the studies by Jones and by Baiz. In Table 5 the OR is reported for a 10 nmol/L increase in the study by Palmer, which is the preferable reporting style.

The reporting of the results from the study by Weisse (Table 3) are slightly confusing as the quartiles include case numbers and not ORs as for the other studies, and it remains unclear if the single OR estimate refers to any of the categories (low vs high or opposite) or a continuous effect.

I suggest changing “positive association” for the study by De Jongh to “Low maternal serum 25(OH)D levels decreased the risk of… ” which is more consistent with the reporting of other studies, and easier to understand.

Other comments

Line 87-88: Reference has been inserted as author/year, but should be a number to be consistent with style.

Title of Tables 4 and 5: suggest replacing "and cordblood" with "or cordblood" as in title of Table 3

The unit of vitamin D concentration (nmol/L) seems to be converted from the original units in the paper in one or more studies (e.g. Weisse 2013 reported in ng/mL) – should be mentioned under Methods if the reported units differ from the original study.

Reviewer #3: It is difficult to grasp the aims and originality of this analysis.

Lines 133-138 detail the aims of the analysis. However, after reading multiple times, I cannot distinguish the difference between aims 2 and 3. No effect is measured, all the studies included are associations.

The authors concluded that there were no associations between 25(OH)D levels during pregnancy and at birth, and 3 respiratory based outcomes in infants up to one year of age. The relevant discussion that address why this may be so has been removed from the document. 25(OH)D is a biomarker of being in the sun but it is not necessarily the active molecule for the outcomes investigated. This important point has been removed. Reasons for a lack of association requires discussion.

Lines 58, 274, 277, Asian countries not Asia countries

Line 58, remove while.

Line 55, prevalence is singular and requires a singular not plural verb.

7. PLOS authors have the option to publish the peer review history of their article (what does this mean?). If published, this will include your full peer review and any attached files.

Reviewer #1: No

Reviewer #3: No

---

## [Author Response · Author response to Decision Letter 2]

7 May 2020

We would like to express our sincere gratitude to the esteemed reviewers for their useful comments. We have now revised the manuscript according to the reviewers’ suggestions, and sincerely hope that the manuscript is now suitable for publication in Plos One. Also, we had sent for proofreading for this revised manuscript.

Below are the author response to the reviewer comments.

Comments from Reviewer 1

1. The authors have done a thorough revision. The study rationale is much now clearer and the manuscript reads a lot better. Most of my comments have been addressed and tables have improved considerably. A few minor points remain regarding the reporting of the results.

Author Response: Thank you for the comment. We had amended it accordingly.

2. Table 2: no apparent order of studies. Could be sorted by publication year.

Author Response: Thank you for the suggestion. We had sorted the information in Table 2 by the recent year of publication accordingly. 

3. The marking of significant ORs in bold does still not seem entirely consistent, e.g. in Table 4 significant ORs are unmarked for the studies by De Jongh (OR for Q2) and Baiz. In Table 5 the same applies to the studies by De Jongh (Q3) and by Camargo.

Author Response: We appreciate the comments and we had bold the information and corrected it accordingly. 

4. It remains unclear for some studies what unit/increase the ORs are reported for, e.g. Table 4 for the studies by Jones and by Baiz. In Table 5 the OR is reported for a 10 nmol/L increase in the study by Palmer, which is the preferable reporting style.

Author Response:We are sorry for the missing information. The information that is requested already been updated in the Table 4.

5. The reporting of the results from the study by Weisse (Table 3) are slightly confusing as the quartiles include case numbers and not ORs as for the other studies, and it remains unclear if the single OR estimate refers to any of the categories (low vs high or opposite) or a continuous effect.

Author Response: Thank you for highlighting this. We did re-checking the reporting result of Weisse et al. (2013) study. The study reported they performed multivariate logistic regression and presented case numbers in each quartiles instead of ORs as for the other studies. As regards to eczema outcome definition, the eczema was recorded as parental report of symptoms and also parental report of a doctor diagnosed. Therefore, there are two results of eczema were presented. 

6. I suggest changing “positive association” for the study by De Jongh to “Low maternal serum 25(OH)D levels decreased the risk of… ” which is more consistent with the reporting of other studies, and easier to understand.

Author Response: Thank you for the suggestion. We had changed the sentence as suggested accordingly. 

7. Line 87-88: Reference has been inserted as author/year, but should be a number to be consistent with style.

Author Response:Thank you for highlighting this. We had amended it accordingly. Line 88.

8. Title of Tables 4 and 5: suggest replacing "and cordblood" with "or cordblood" as in title of Table 3

Author Response: Thank you for the suggestion. We had changed the phrase as suggested in title of Table 4 and 5 accordingly. 

9. The unit of vitamin D concentration (nmol/L) seems to be converted from the original units in the paper in one or more studies (e.g. Weisse 2013 reported in ng/mL) – should be mentioned under Methods if the reported units differ from the original study.

Author Response: Thank you for highlighting this. After further discussion among the authors, we had changed the converted ones to the original units in the papers. However, for clarity purposes for comparison (the ng/ml change to nmol/L), we put a footnote, 1 unit ng/ml equivalent to 2.496 nmol/L. The information already been updated in footnote each table accordingly.

Comments from Reviewer 3

1. It is difficult to grasp the aims and originality of this analysis. Lines 133-138 detail the aims of the analysis. However, after reading multiple times, I cannot distinguish the difference between aims 2 and 3. No effect is measured, all the studies included are associations.

Author Response: We are sorry for the confusion of the information. To avoid the misunderstanding, we had removed the goal of no.3 as our review summarized and elaborated the findings in a narrative manner.

2. The authors concluded that there were no associations between 25(OH)D levels during pregnancy and at birth, and 3 respiratory based outcomes in infants up to one year of age. The relevant discussion that address why this may be so has been removed from the document. 25(OH)D is a biomarker of being in the sun but it is not necessarily the active molecule for the outcomes investigated. This important point has been removed. Reasons for a lack of association requires discussion.

Author Response: Thank you for highlighting this. The information that is requested already been updated accordingly. Line 288-291.

3. Lines 58, 274, 277, Asian countries not Asia countries

Author Response: Thank you for highlighting this. We had amended it accordingly. Line 59, 277, 286, 289

4. Line 58, remove while.

Author Response: Thank you for highlighting this. We had removed ‘while’ at line 58.

5. Line 55, prevalence is singular and requires a singular not plural verb.

Author Response: Thank you for highlighting this. We had amended ‘were most common…’ to ‘was most common..’. Line 55.

---

## [Editor Report · Decision Letter 3]

15 May 2020

Risk of eczema, wheezing and respiratory tract infections in the first year of life: a systematic review of vitamin D concentrations during pregnancy and at birth

PONE-D-19-24730R3

Dear Dr. Majid,

We are pleased to inform you that your manuscript has been judged scientifically suitable for publication and will be formally accepted for publication once it complies with all outstanding technical requirements.

With kind regards,

Maria Christine Magnus, MPH

Academic Editor

PLOS ONE
---

## [Editor Report · Acceptance letter]

3 Jun 2020

PONE-D-19-24730R3 

Risk of eczema, wheezing and respiratory tract infections in the first year of life: a systematic review of vitamin D concentrations during pregnancy and at birth 

Dear Dr. Majid:

I'm pleased to inform you that your manuscript has been deemed suitable for publication in PLOS ONE. Congratulations! Your manuscript is now with our production department. 

Kind regards, 

on behalf of

Dr. Maria Christine Magnus 

Academic Editor

PLOS ONE